# Allele-specific endogenous tagging and quantitative analysis of β-catenin in colorectal cancer cells

Giulia Ambrosi[1†], Oksana Voloshanenko[1†], Antonia F Eckert[2†], Dominique Kranz[1], G Ulrich Nienhaus[2,3,4,5]*, Michael Boutros[1]*

[1]German Cancer Research Center (DKFZ), Division of Signaling and Functional Genomics and Heidelberg University, BioQuant and Medical Faculty Mannheim, Heidelberg, Germany; [2]Institute of Applied Physics, Karlsruhe Institute of Technology, Karlsruhe, Germany; [3]Institute of Nanotechnology, Karlsruhe Institute of Technology, Karlsruhe, Germany; [4]Institute of Biological and Chemical Systems, Karlsruhe Institute of Technology, Karlsruhe, Germany; [5]Department of Physics, University of Illinois at Urbana-Champaign, Urbana, United States

*For correspondence:
uli@uiuc.edu (GUN);
m.boutros@dkfz.de (MB)

[†] These authors contributed equally to this work

Competing interest: The authors declare that no competing interests exist.

**Abstract** Wnt signaling plays important roles in development, homeostasis, and tumorigenesis. Mutations in β-catenin that activate Wnt signaling have been found in colorectal and hepatocellular carcinomas. However, the dynamics of wild-type and mutant forms of β-catenin are not fully understood. Here, we genome-engineered fluorescently tagged alleles of endogenous β-catenin in a colorectal cancer cell line. Wild-type and oncogenic mutant alleles were tagged with different fluorescent proteins, enabling the analysis of both variants in the same cell. We analyzed the properties of both β-catenin alleles using immunoprecipitation, immunofluorescence, and fluorescence correlation spectroscopy approaches, revealing distinctly different biophysical properties. In addition, activation of Wnt signaling by treatment with a GSK3β inhibitor or a truncating *APC* mutation modulated the wild-type allele to mimic the properties of the mutant β-catenin allele. The one-step tagging strategy demonstrates how genome engineering can be employed for the parallel functional analysis of different genetic variants.

## Editor's evaluation

The authors describe a fluorescent tagging strategy that allows them to knock in one copy each of differently tagged wild-type and a non-destructible β-catenin mutant, so that the behavior of the mutant can be directly compared to the wild-type protein under different activities of the β-catenin destruction complex. The differential tagging of a wild-type and mutant Wnt components offers the opportunity to compare directly their properties in a living cell, and the approach is likely to be useful for future mechanistic investigations by the Wnt and cancer communities.

## Introduction

Wnt signaling pathways play fundamental roles in many biological processes in development, stem cell maintenance, and disease (*Clevers and Nusse, 2012*; *Zhan et al., 2017*). A key regulator of "canonical" Wnt signaling is β-catenin (*CTNNB1*). When Wnt signaling is inactive, β-catenin forms part of the destruction complex, which consists of adenomatous polyposis coli (APC), axis inhibition protein 1 (Axin1) (*Li et al., 2012*), casein kinase 1α (CK1α), and glycogen synthase kinase 3β (GSK3β). In the destruction complex, β-catenin is N-terminally phosphorylated at Ser45 by CK1 kinase (*Hagen*

*and Vidal-Puig, 2002*) and then sequentially phosphorylated at residues Ser33, Ser37, and Thr41 by GSK3β (*Liu et al., 2002*). Subsequently, phospho-β-catenin is targeted to proteasomal degradation by the SCF^βTrCP E3-ligase complex (*Wu et al., 2003*).

Wnt signaling is activated when Wnt ligands bind to seven-pass transmembrane receptors of the Frizzled (Fzd) protein family and the transmembrane LRP5/6 (low-density lipoprotein receptor-related protein) co-receptor (*Bilic et al., 2007*; *Cong et al., 2004*). Fzd proteins then recruit the cytosolic adaptor protein Dishevelled (Dvl) to the cell membrane *via* its DEP domain, leading to polymerization of Dvl (*Schwarz-Romond et al., 2007*). Dvl facilitates membrane recruitment of Axin1 by a DIX-DIX domain heterotypic interaction (*Cliffe et al., 2003*), which in turn stimulates GSK3β- and CK1α-mediated phosphorylation of the LRP5/6 cytosolic tail in its five PPPSPxS motifs and its binding to LRP5/6 (*Stamos et al., 2014*). As a consequence, the destruction complex disassembles, β-catenin is no longer degraded and translocates into the nucleus. In the nucleus, β-catenin interacts with the T cell factor (TCF)/lymphoid enhancer-binding factor (LEF) family of transcription factors and activates target genes in a cell-type-dependent manner. In addition to its role in Wnt signaling, β-catenin is also found at cell-cell adherens junctions (AJs) mediating the interactions between the cytoplasmic domain of cadherins and the actin cytoskeleton (*Aberle et al., 1994*; *Hoschuetzky et al., 1994*; *Huber and Weis, 2001*).

The human β-catenin protein consists of 781 amino acids and contains three distinct structural domains. Crystal structure and NMR analysis of β-catenin have revealed that both N- and C-terminal domains are structurally flexible, whereas the central armadillo repeat has a rather rigid scaffold structure (*Huber et al., 1997*; *Orsulic and Peifer, 1996*; *Xing et al., 2008*). The N-terminus is crucial for its stability and for cell adhesion by interacting with α-catenin. This region also contains evolutionarily conserved serine and tyrosine residues (*Valenta et al., 2012*). Dysregulation of Wnt signaling components is associated with a variety of human diseases, ranging from growth-related pathologies to cancer (*Clevers and Nusse, 2012*; *Zhan et al., 2017*); however, overactive Wnt signaling has been difficult to target pharmacologically (*Zhong and Virshup, 2020*).

Mutations in *CTNNB1*/β-catenin in colorectal cancer are typically found in its N-terminal domain, particularly in exon 3 that carries multiple phosphorylation sites for CK1 and GSK3β, including amino acids Ser33, Ser37, Thr41, and Ser45 (*Cancer Genome Atlas Network, 2012*; *Dar et al., 2017*; *Jamieson et al., 2011*). Mutations or deletions of these sites, often in only one allele, prevent phosphorylation of β-catenin, leading to its accumulation and subsequent activation of Wnt pathway target genes. However, the biochemical and biophysical properties of different β-catenin alleles in their endogenous loci have remained largely unknown.

To functionally analyze proteins, the introduction of small immune epitopes or fluorescent tags in-frame of the genomic locus enables their visualization and quantitative biophysical analysis. In this study, we aimed to examine the behavior of wild-type and a mutant oncogenic β-catenin. We made use of the colon cancer cell line HCT116, which harbors one wild-type and one mutant ΔSer45 β-catenin allele, to generate endogenously tagged β-catenin by CRISPR/Cas9 genome editing. We engineered an HCT116 clone with mClover-tagged wild-type β-catenin and mCherry-tagged mutant β-catenin. Tagging wild-type and mutant β-catenin in the same cell provided an opportunity to compare the function and behavior of both proteins in parallel in the same sample. Our results indicate that wild-type and mutant β-catenin exist in two separate pools differing in their physical properties. Moreover, treatment with a GSK3β inhibitor or introduction of a truncating mutation of *APC* changed the physical properties of wild-type β-catenin, phenocopying the ΔSer45 mutant isoform.

## Results
### A strategy to generate endogenously tagged fluorescent β-catenin isoforms

Previously, the dynamics and biochemical properties of β-catenin in cells were investigated by using transiently or stably overexpressed fluorescent fusion proteins (*Giannini et al., 2000*; *Jamieson et al., 2011*; *Kafri et al., 2016*; *Krieghoff et al., 2006*). To study wild-type and mutant β-catenin at physiological expression levels, we utilized the diploid colon cancer cell line HCT116 that harbors both a wild-type and a ΔSer45 mutant allele. This in-frame ΔSer45 deletion results in a loss of phosphorylation

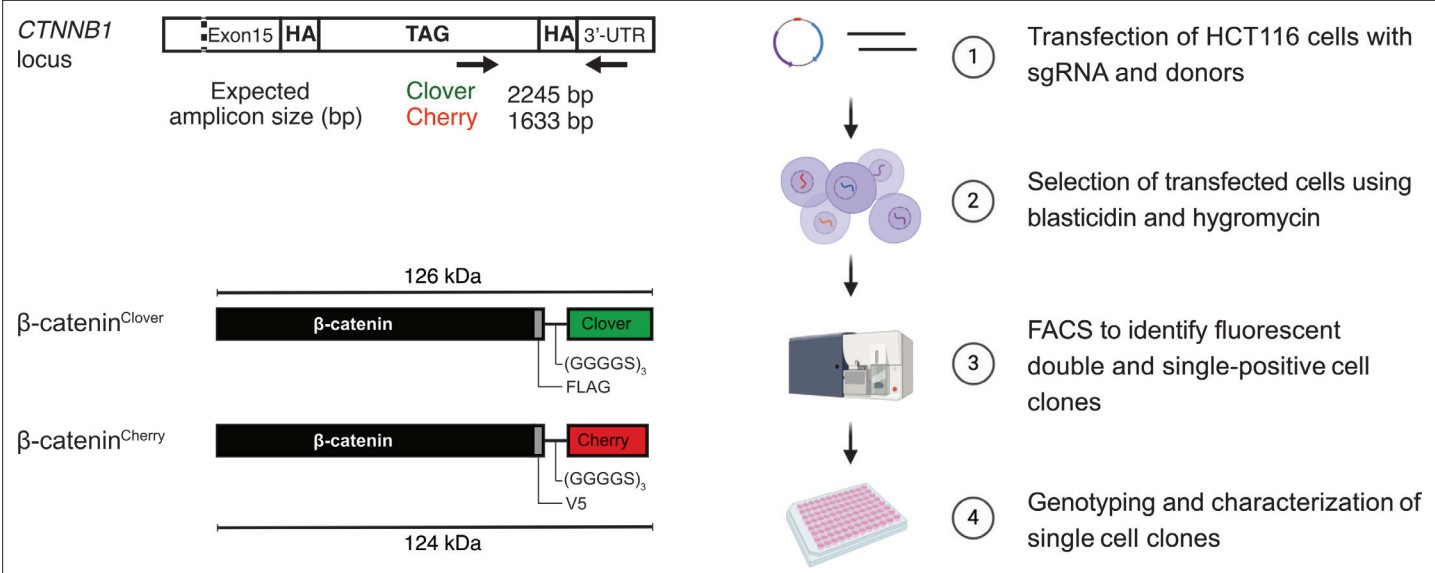

**Figure 1.** Strategy and workflow for bi-allelic fluorescent tagging of β-catenin in colon cancer cells. (Left panels) Schematic representation of *CTNNB1*/ β-catenin tagging strategy, the CTNNB1/β-catenin locus and tagged β-catenin proteins. (Right panels) Workflow for the generation of endogenously tagged β-catenin in HCT116 colorectal cancer cells. See also *Figure 1—figure supplement 1A* for more details. bp, base pair; HA, homology arm; kDa, kilo Dalton; sgRNA, single guide RNA; UTR, untranslated region.

The online version of this article includes the following figure supplement(s) for figure 1:

**Figure supplement 1.** Generation of endogenous fluorescently tagged β-catenin cell lines.

**Figure supplement 2.** Characterization of endogenous fluorescently tagged β-catenin cell lines.

**Figure supplement 3.** The sequences of integration sites/tags of bi-allelic β-catenin tagged HCT116 cells (#37) in comparison to donor templates.

**Figure supplement 4.** The sequences of integration sites/tags of bi-allelic β-catenin tagged HCT116 cells (#37) in comparison to donor templates.

at this position, leading to the stabilization of β-catenin and constant activation of Wnt downstream targets (*Hagen and Vidal-Puig, 2002*).

To generate endogenously tagged β-catenin, we applied a gene replacement strategy using CRISPR/Cas9 editing (*Mali et al., 2013*; *Ran et al., 2013*; *Yang et al., 2014*) to introduce different fluorescent tags in the two alleles of the β-catenin locus of HCT116 cells in one step (*Figure 1*, *Figure 1—figure supplement 1A*). To this end, we identified a suitable sgRNA sequence for targeting the C-terminal region of β-catenin. This target site was chosen because it has previously been shown that C-terminal tagging did not affect β-catenin function in overexpression experiments (*Giannini et al., 2000*; *Jamieson et al., 2011*; *Kafri et al., 2016*; *Krieghoff et al., 2006*). Then, two donor templates with 180 bp homology arms were designed close to the sgRNA PAM sequence (*Elliott et al., 1998*; *Natsume et al., 2016*) encoding two different fluorophores. We chose mClover3 and mCherry2 for their favorable biophysical properties such as spectral absorbance, brightness, photostability, and maturation time (*Bajar et al., 2016*; *Shen et al., 2017*; *Thorn, 2017*). In the following, for the sake of brevity, we will refer to these two fluorescent proteins as Clover and Cherry, respectively. Moreover, FLAG and V5-tags were integrated between the C-terminal domain of β-catenin and the flexible GS linker to enable biochemical experiments. To avoid steric hindrance of the different domains, a flexible GS linker consisting of three repeats of glycine and serine residues, (Gly-Gly-Gly-Gly-Ser)$_3$, was inserted between the FLAG/V5 tags and the fluorescent proteins (*Trinh et al., 2004*; *Figure 1*, *Figure 1—figure supplement 1A*). Gene resistance cassettes flanked by loxP sites were also introduced after the fluorescent proteins for selection of edited cells after homology-directed repair (HDR) (*Smirnikhina et al., 2019*). Selection cassettes encoded for antibiotic resistance to hygromycin and blasticidin on the Clover and Cherry donor templates, respectively (*Figure 1—figure supplement 1A*). The loxP sites were included to allow resistance cassette removal upon expression of Cre recombinase.

## Generation and characterization of β-catenin tagged cells

To generate cell lines with endogenously allele-specific tagged β-catenin, we transfected HCT116 cells with plasmids encoding for both the sgRNA targeting β-catenin (sgCTNNB1) and Cas9 protein from *Streptococcus pyogenes* (*Mali et al., 2013*) and the donor template encoding either V5-mCherry BRS (blasticidinS) or FLAG-mClover HygR (hygromycin resistance) (*Figure 1*, *Figure 1—figure supplement 1A*). Transfected HCT116 cells were first selected with puromycin for 48 hr and subsequently with blasticidin/hygromycin for 5 days. Pooled edited cells were analyzed and sorted into single-cell clones by fluorescence-activated cell sorting (FACS) (*Figure 1*, *Figure 1—figure supplement 1B*). Infrequent fluorophore expression was observed, regardless of whether a single donor template was used (1.4% of Cherry- and 17.3% of Clover-labeled cells) or both donors were co-transfected simultaneously (0.9% of only Cherry-, 7.4% of only Clover-, and 0.1% of double-labeled cells) (*Figure 1—figure supplement 1B*).

In total, 44 single-cell clones were sorted by FACS and expanded for further characterization, which included double-positive and single-positive cells (*Figure 1—figure supplement 1C*). All 44 clones were analyzed by allele-specific PCR and most of them were sequenced from the 5′ junction of the integration cassette. Only one single clone (#37) appeared to have the correct insertion of both fluorophores in the β-catenin/*CTNNB1* locus. In addition, 7 clones showed a heterozygous integration of the Cherry fluorophore in the β-catenin/*CTNNB1* locus, and 24 clones were heterozygous for the Clover fluorophore-tagged allele. Moreover, six clones had an insertion of the donor cassette after the stop codon of β-catenin/*CTNNB1*. Two isogenic cell lines had partial integration of the Clover tag and four clones were wild-type indicating no editing events in the *CTNNB1* locus (*Figure 1—figure supplement 1C*). Presumably, they were selected due to random integration or high background autofluorescence of these clones.

In the following, we focus on clones #24, #33, #37, and #45, as they were found to represent different integration events. To confirm the editing events, genomic DNA, mRNA, and protein lysates were analyzed and compared to the parental HCT116 (β-catenin$^{WT/\Delta45}$) cell line and to an HCT116 isogenic cell line in which the ΔSer45 allele was removed (β-catenin$^{WT+/−}$) (*Chan et al., 2002*). To analyze them in depth, we used primers annealing upstream and downstream of the fluorophore tags (primers (a) in *Figure 1—figure supplement 2A*). While PCR amplicons of successfully tagged *CTNNB1*/β-catenin alleles are expected to be more than 3 kb in length, amplicons from the untagged locus would be approximately 1.2 kb. PCR from genomic DNA from clone #37 resulted in a single band of 3 kb, indicating that both alleles of *CTNNB1* were edited. In contrast, genotyping of clones #33, #45, and #24 resulted in 3 kb and 1.2 kb amplicons, suggesting that only a single donor integrated in one allele (*Figure 1—figure supplement 2A*). Since genotyping with primers annealing within the donor construct cannot distinguish between homology-mediated integration at the target locus and non-targeted integration elsewhere in the genome, we next performed genotyping PCRs with one primer annealing in the *CTNNB1* locus and the other one inside the integration cassette (*Figure 1—figure supplement 2A*). Separate PCRs were done for the 5′ and 3′ junction of the integration cassette; each confirmed correct integration at the target locus in all four clones. We also used primers that anneal selectively to either the Cherry or Clover donor construct or can be used to distinguish between these two donors by the size of their resistance gene (primers (b), (b′), (c), (e), and (e′) in *Figure 1—figure supplement 2A*). These PCRs revealed that clones #33, #37, and #24 harbor the Clover tag, while clones #37 and #45 have the Cherry tag integrated at the *CTNNB1* locus (*Figure 1—figure supplement 2A*). Finally, we performed PCR with a primer pair annealing at either side of the donor cassette, with the resulting amplicon spanning the entire inserted exogenous sequence. PCR of genomic DNA from clone #37 gave rise to a single band of the expected size of approximately 3 kb, indicating correct insertion of the donor cassette in both alleles (*Figure 1—figure supplement 2A*). Taken together, these experiments indicate that single-cell clones #24 and #33 had the Clover donor construct inserted in one *CTNNB1* allele, clone #45 had the Cherry tag inserted in one allele, and clone #37 had one allele tagged with Cherry and the other one tagged with Clover.

Next, we sequenced the amplicons from our genotyping PCRs (*Figure 1—figure supplements 3 and 4*). This revealed correct in-frame integration of the fluorophore sequences at the 3′ end of the CTNNB1/β-catenin coding sequence. We also confirmed that one *CTNNB1* allele remained untagged in clones #24, #33, and #45. While the untagged allele was not altered at the sgRNA target site in clones #24 and #45, we detected a 40-bp mutation in clone #33, presumably from error-prone

non-homologous end joining of a Cas9-induced double-strand break. This mutation results in four additional amino acids at the end of β-catenin. Importantly, sequencing of the bi-allelically tagged clone #37 revealed that both alleles had been modified without unintended mutations (*Figure 1—figure supplements 3 and 4*).

Subsequently, PCR was performed to determine which allele (wild-type or ΔSer45) was tagged with Clover and which with Cherry using cDNA from reverse-transcribed mRNA of each isogenic cell line and specific reverse primers to Clover or Cherry. Sanger sequencing of cDNA templates revealed that both clones #37 and #45 carried the Cherry fluorophore in the ΔSer45 allele (*Figure 2A*). In contrast, clones #33 and #37 harbored the Clover fluorophore in the wild-type allele, as indicated by the in-frame sequence with Ser45.

Next, total cell lysates of parental and isogenic knock-in cell lines were analyzed for β-catenin fusion proteins by Western blot (*Figure 2B*, *Figure 2—figure supplement 1A*). As shown in *Figure 2B*, clones #33 and #45 have one of the tagged variants of β-catenin, whereas clone #37 has two high molecular weight bands indicative of both tagged β-catenin variants. Furthermore, immunoprecipitations using either affinity beads for the Clover (GFP) and Cherry (RFP) fluorophores, or anti-β-catenin antibodies were performed and analyzed by Western blotting. GFP-immunoprecipitated lysates of clone #37 (*Figure 2C*), a β-catenin variant with molecular mass higher than 97 kDa was recognized by β-catenin, GFP, and Flag antibodies. By contrast, upon RFP/Cherry immunoprecipitation a band around 97 kDa was detected by β-catenin, Cherry and V5 antibodies. Pulldown with a β-catenin antibody demonstrated two different β-catenin variants that were either detected by GFP/Flag or Cherry/V5 antibodies. These results confirm the presence of two differently tagged β-catenin variants in clone #37 at the protein level. Similarly, we analyzed clones #33 and clone #45 by immunoprecipitation (*Figure 2—figure supplement 1B,C*). In clone #33, GFP-immunoprecipitated lysates showed a β-catenin variant higher than 97 kDa. By contrast, in clone #45 upon Cherry-immunoprecipitation β-catenin, Cherry and V5 antibodies recognized bands around 97 kDa.

Wild-type β-catenin-Clover protein runs higher than 97 kDa and it is recognized by both GFP and Flag antibodies (*Figure 2B and C*, *Figure 2—figure supplement 1*). Mutant β-catenin-Cherry protein runs around 97 kDa as a single band, as detected in the whole-cell lysate (*Figure 2B and C*). Upon immunoprecipitation, especially with RFP beads, several bands of mutant β-catenin-Cherry are frequently detected. To evaluate whether this is due to modifications of mutant β-catenin or due to the Cherry-tag itself, we analyzed clone #24 expressing a mutant β-catenin tagged with Clover. All three antibodies, β-catenin, GFP, and Flag, detected only one band representing the mutant β-catenin-Clover protein (*Figure 2—figure supplement 1D*), suggesting that the additional bands of mutant β-catenin-Cherry arise from the Cherry-tag. One possible explanation is that the mCherry protein undergoes chemical modification during sample preparation, for example, by TCEP in the loading dye (*Cloin et al., 2017*). Alternatively, mCherry might undergo backbone cleavage during chromophore maturation (*Barondeau et al., 2006*; *Nienhaus et al., 2005*; *Wei et al., 2015*). Importantly, such cleavage during maturation does not impair the fluorescence emission, as fragments remain tightly associated to form the fluorescent beta-barrel structure (*Barondeau et al., 2006*; *Nienhaus et al., 2005*).

## Functional characterization of endogenously tagged β-catenin cell lines

To test whether the C-terminal fluorescent tags of β-catenin interfere with its physiological functions, we tested the novel cell lines in different assays. First, we compared cell proliferation of the clonal cell lines to the parental β-catenin[WT/Δ45] and the β-catenin[WT+/−] cell line, which only had a wild-type but not a mutant β-catenin allele. All three clones #33, #37, and #45 showed a similar growth behavior as the parental β-catenin[WT/Δ45] cells (*Figure 3A*), whereas the cell line carrying a single wild-type allele (β-catenin[WT+/−]) displayed impaired growth. Next, we tested the functionality and localization of the endogenously tagged β-catenin variants. We performed siRNA-mediated silencing of β-catenin and analyzed gene expression by RT-qPCR (*Figure 3B*, *Figure 3—figure supplement 1A*). We found that downregulation of β-catenin led to the reduction in fluorophore levels as well as the Wnt target gene *AXIN2*. Consistently, depletion of β-catenin resulted in a reduction of Cherry and Clover, as shown by immunofluorescence on the protein level (*Figure 3C*, *Figure 3—figure supplement 1A*).

In addition to its functional role in Wnt signaling, β-catenin interacts with the cytoplasmic domains of E-cadherin and α-catenin to bridge the cell-adhesion complex and actin cytoskeleton inside the

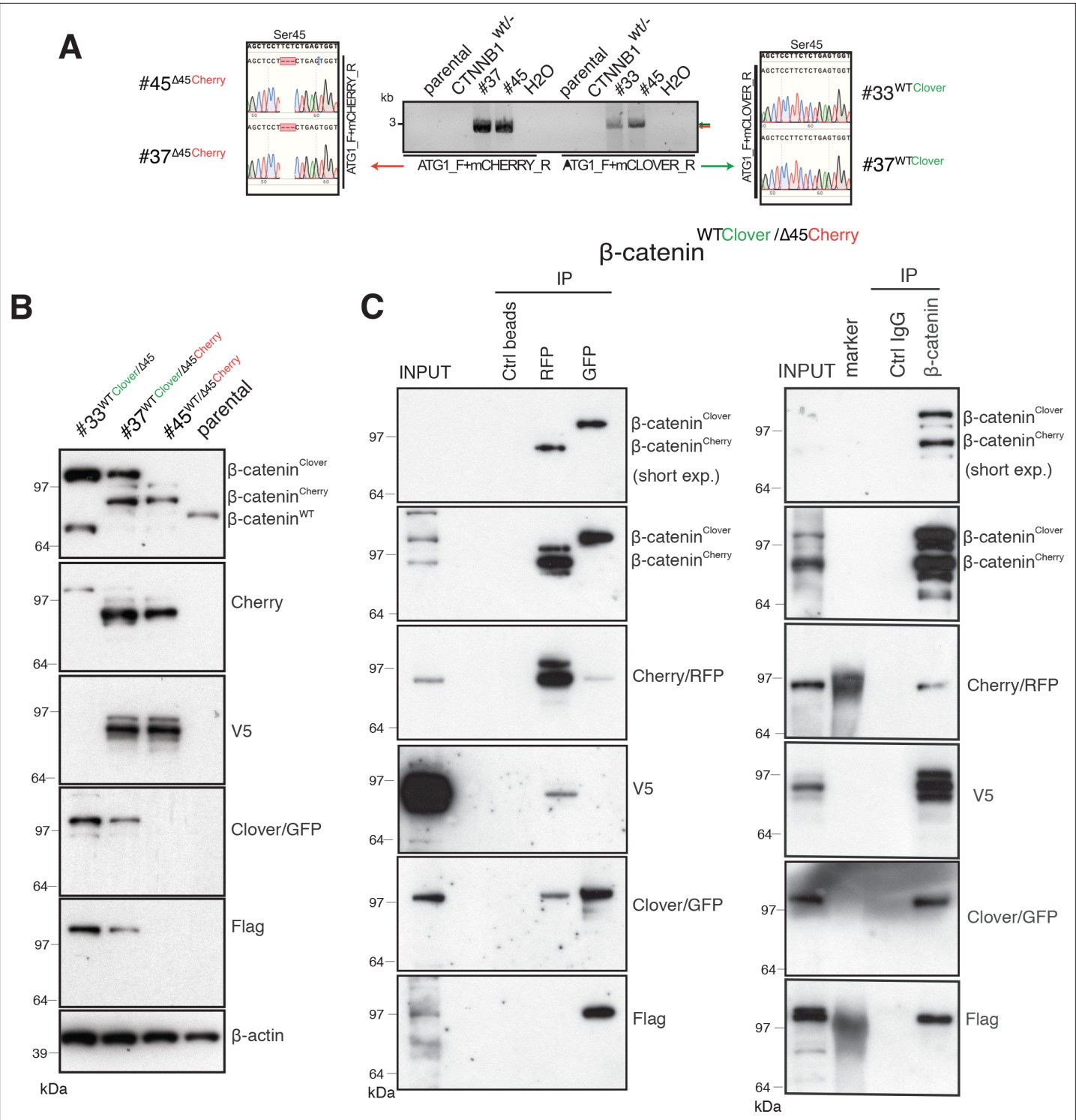

**Figure 2.** Identification and confirmation of tagged β-catenin alleles. (**A**) Sanger sequencing confirms bi-allelic tagging of β-catenin. Sequencing results show clones #45 (β-catenin$^{WT/Δ45Cherry}$) and #37 (β-catenin$^{WTClover/Δ45Cherry}$) Cherry is in-frame with the mutant allele and clones #33 (β-catenin$^{WTClover/Δ45}$) and #37 (β-catenin$^{WTClover/Δ45Cherry}$) Clover is in-frame with wild-type allele (codon TCT). (**B**) Cell lysates of indicated HCT116 cell lines analyzed by Western blotting with a β-catenin antibody; β-actin served as a loading control. (**C**) HCT116 β-catenin$^{WTClover/Δ45Cherry}$ (clone #37) immunoprecipitation with GFP, Cherry and control beads, or with a β-catenin antibody followed by immunoblotting with indicated antibodies. Representative results from three independent experiments are shown.

The online version of this article includes the following source data and figure supplement(s) for figure 2:

*Figure 2 continued on next page*

*Figure 2 continued*

**Source data 1.** Identification and confirmation of tagged β-catenin alleles.

**Figure supplement 1.** Validation of endogenously fluorescent-tagged β-catenin in HCT116 colon cancer cells.

cells, thereby localizing it to the cell membrane (*Aberle et al., 1994*; *Hoschuetzky et al., 1994*). Immunofluorescence analysis confirmed the localization of the endogenously tagged β-catenin variants at the cell membrane as observed in the parental HCT116 cell line (*Figure 3D*, *Figure 3—figure supplement 1B*). Immunoprecipitation with an E-cadherin antibody demonstrated the interaction of β-catenin and E-cadherin in all clones (*Figure 3E*, *Figure 3—figure supplement 1C*). Taken together, these results indicate that endogenously tagging of β-catenin did neither affect its localization and adhesive function nor its ability to activate the Wnt pathway.

## Wild-type and mutant β-catenin variants contribute both to canonical Wnt signaling output

We next investigated the impact of endogenously tagged β-catenin on the activation of Wnt signaling. Wnt activity was measured by TCF4/Wnt-dependent reporter assay and analyzing *AXIN2* levels in the isogenic knock-in clones and parental cell line upon treatment with either the GSK3β inhibitor CHIR99021 or ICG001, an inhibitor of CBP/β-catenin interaction (*Figure 4A*). Upon CHIR99021 treatment, TCF4/Wnt-reporter activity was induced and *AXIN2* gene expression levels were increased similarly in all isogenic cell lines. Treatment with the Wnt inhibitor ICG001 decreased Wnt reporter activity and *AXIN2* expression levels in all cell lines, as expected (*Figure 4A*). In addition, immunofluorescence analysis demonstrated that after addition of CHIR99021, nuclear β-catenin was increased in all clones indicative of increased nuclear translocation of β-catenin upon Wnt activation (*Figure 4B*, *Figure 4—figure supplements 1 and 2*). Consistent with its mechanism of action, treatment with ICG001 did neither affect cytoplasmic nor nuclear β-catenin localization in the endogenously tagged or parental cell clones. Upon inhibition of Wnt secretion by addition of porcupine inhibitor LGK974, the signal of both, mutant as well as wild-type β-catenin allele, decreased but could be rescued by addition of recombinant Wnt3a (*Figure 4C*).

To analyze the impact of both wild-type and mutant variants on Wnt signaling, we performed knockdown experiments with siRNAs targeting either Cherry or Clover, thereby depleting either mutant or wild-type β-catenin alleles. Silencing of either Clover or Cherry in clone #37 reduced β-catenin and *AXIN2* expression levels in a similar manner, whereas combined knockdown of both had an additive effect comparable to siRNA β-catenin/*CTNNB1* (*Figure 5A*). Results of mRNA expression are in line with the immunofluorescence staining. Silencing of Cherry or Clover led to a decrease in signal intensity of the corresponding β-catenin alleles (*Figure 5B*, *Figure 5—figure supplement 1*). These results indicate that both wild-type and mutant β-catenin alleles contribute to Wnt activity. This is also consistent with previous results in HCT116 cells demonstrating that Wnt secretion is essential for maintaining Wnt activity (*Voloshanenko et al., 2013*).

## Fluorescence correlation spectroscopy reveals differences in the dynamics of wild-type and mutant β-catenin

Fluorescence correlation spectroscopy (FCS) is a powerful quantitative microscopy method for measuring local concentrations and diffusional mobilities of fluorescent molecules in cells and tissues, for example, to determine Wnt ligand-receptor interactions in the cell membrane (*Eckert et al., 2020*). To simultaneously monitor the dynamics of wild-type Clover-tagged and mutant Cherry-tagged β-catenin of HCT116 cells clone #37, we used FCS with detection in two color channels. With this method, we recorded the fluorescence intensity emanating from a tiny observation volume (ca. 1 fL) as a function of time. For each cell studied, two measurements were performed, with the observation volume first positioned into the cytosol and then into the nucleus. Thus, we focused on the freely diffusing β-catenin fraction, which mediates Wnt signaling, and excluded the membrane-bound fraction, which is responsible for cell-cell adhesion and has been previously shown by FRAP to tightly interact with cadherins, which markedly reduces its mobility (*Krieghoff et al., 2006*).

From the intensity time traces in the two color channels, autocorrelation functions were calculated and fitted with model functions (*Figure 6—figure supplement 1A*) to extract two parameters, the

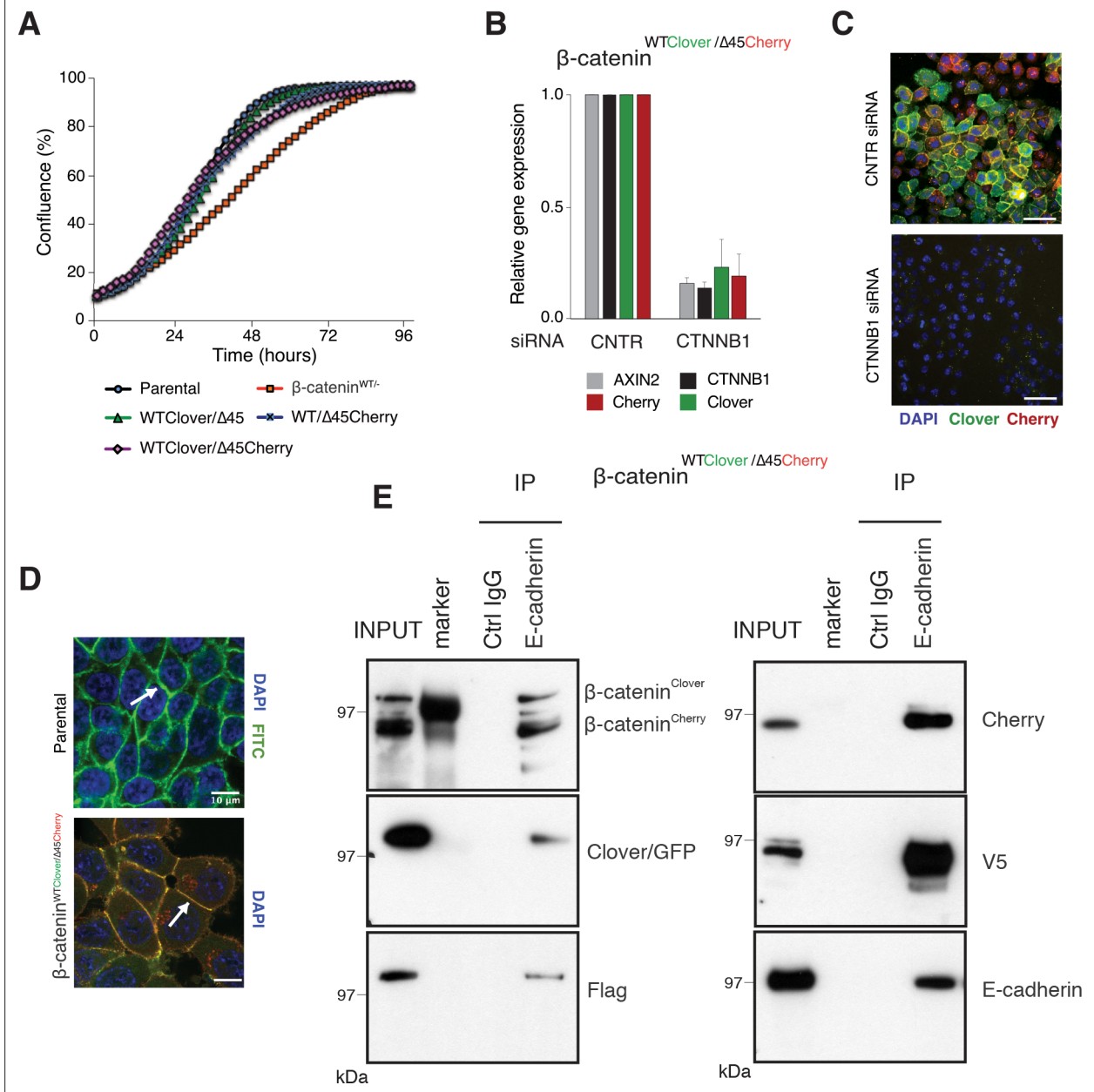

**Figure 3.** Fluorescently tagged β-catenin variants are functional and localize to adherens junctions. (**A**) Proliferation of the indicated HCT116 cell clones was monitored by live-cell imaging using an Incucyte instrument. (**B**) In HCT116 β-catenin[WTClover/Δ45Cherry], mRNA-levels of *CTNNB1*, *AXIN2*, *Cherry,* and *Clover* were determined by RT-qPCR in control conditions and upon depletion of *CTNNB1*/β-catenin by siRNA (n=5; mean ± SD). (**C**) Representative immunofluorescence images of HCT116 β-catenin[WTClover/Δ45Cherry] after siRNA-mediated knockdown of *CTNNB1* are shown (n=3; scale bars: 100 μm). (**D**) β-catenin accumulates at cell-cell junctions (arrow). Representative immunofluorescence images of HCT116 β-catenin[WTClover/Δ45Cherry] and parental HCT116 β-catenin[WT/Δ45] stained with a β-catenin antibody are shown (scale bars: 10 μm). (**E**) Immunoprecipitation of HCT116 clone β-catenin[WTClover/Δ45Cherry] with E-cadherin confirms its interaction with β-catenin. Representative results from three independent experiments are shown.

The online version of this article includes the following source data and figure supplement(s) for figure 3:

**Source data 1.** Fluorescently tagged β-catenin variants are functional and localize to adherens junctions.

**Source data 2.** Fluorescently tagged β-catenin variants are functional and localize to adherens junctions.

**Figure supplement 1.** Validation of the physiological function of fluorescently tagged β-catenin.

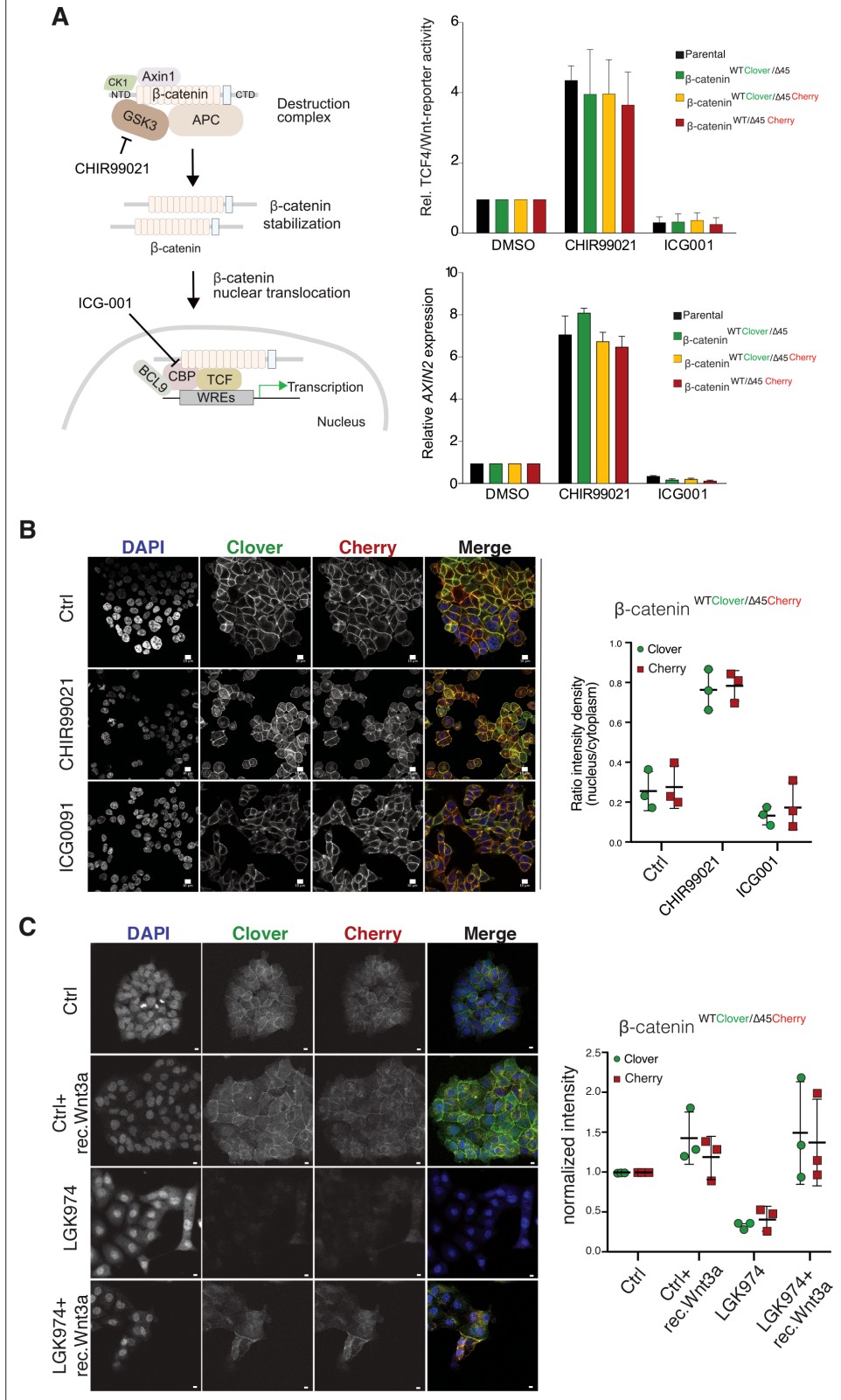

**Figure 4.** Tagging of β-catenin does not affect functionality in canonical Wnt signaling. (**A**) Left: Scheme showing the mode of action of GSK3β inhibitor CHIR99021 and CBP inhibitor ICG-001. Right: Indicated HCT116 cell lines were treated with 10 µM CHIR99021 and 10 µM ICG-001 for 24 hr, then Wnt activity was determined by a luciferase-based TCF4/Wnt-reporter assay (upper panel) and quantification of *AXIN2* mRNA-levels by RT-qPCR (n=3; mean ±

*Figure 4 continued on next page*

*Figure 4 continued*

SD). (**B**) Immunofluorescence analysis of HCT116 β-catenin^WTClover/Δ45Cherry after 24 hr treatment with 10 µM CHIR99021 and 10 µM ICG-001 is shown (scale bar: 10 µm). The graph on the right depicts the ratio of nuclear to cytoplasmic fluorescent signal intensity for Clover and Cherry in HCT116 β-catenin^WTClover/Δ45Cherry. Data from three independent experiments, each with at least 250 cells per condition, are shown as mean ± SEM. Every experiment includes at least 250 cells per condition. Enlarged representative images are shown in the *Figure 4—figure supplement 2*. (**C**) HCT116 β-catenin^WTClover/Δ45Cherry were treated with LGK974 for 80 hr then 200 ng/ml of recombinant Wnt3a was added for 16 hr. Intensities of the Clover or Cherry signals were measured per slide and normalized to the intensity of Hoechst staining and to the control. 10–20 slides were measured per condition in one experiment. Data of three independent experiments are shown as mean ± SEM. Each dot represents an independent experiment. WRE, Wnt responsive element. Scale bar: 10 µm.

The online version of this article includes the following source data and figure supplement(s) for figure 4:

**Source data 1.** Tagging of β-catenin does not affect functionality in canonical Wnt signaling.

**Figure supplement 1.** Tagging of β-catenin does not affect its functionality in canonical Wnt signaling.

**Figure supplement 2.** Tagging of β-catenin does not affect its functionality in canonical Wnt signaling.

diffusional correlation time, and the correlation amplitude (*Zemanová et al., 2003*). The correlation time is inversely proportional to the diffusion coefficient, D, of the fluorescent molecules. For this conversion, we performed calibration FCS experiments using fluorescent dye molecules with known D values as reference samples (see Materials and methods). According to the Stokes-Einstein law, the diffusion coefficient is inversely related to the linear extension of the diffusing entity and thus to the cube root of its volume/molecular mass. Thus, measurement of D values provides size information. The correlation amplitude is inversely proportional to the average number of molecules, N, in the observation volume. The calibration experiment with dye molecules mentioned above also allows us to determine the effective volume, $V_{eff}$, from which the fluorescence emanates, so that the molecule concentration, c, in the observation volume can simply be calculated as $c=N/_{Veff}$. In addition to autocorrelation analysis, we calculated cross-correlation functions to identify correlated intensity fluctuations in the two color channels. These will appear if the Clover and Cherry labeled β-catenin molecules co-diffuse, that is, bind to each other and migrate in concert through the observation volume. Cross-correlation amplitudes between wild-type Clover-tagged and mutant Cherry-tagged β-catenin were always found to be zero within the experimental error, implying that these two β-catenin variants diffuse independently and do not associate to any significant degree (*Figure 6—figure supplement 1A*).

On the basis of FCS experiments in the cytosol of more than 40 cells, we calculated a median D value of 10.6 ± 1.8 µm² s⁻¹ for mutant Cherry-tagged β-catenin, which is significantly smaller than the one of wild-type Clover-tagged β-catenin (D=17.3 ± 6.1 µm² s⁻¹, *Figure 6A*). The ratio of diffusion coefficients indicates that mutant Cherry-tagged β-catenin diffuses as part of a larger complex, with a more than fourfold larger volume/overall mass than wild-type Clover-tagged β-catenin. For Clover and Cherry overexpressed in HCT116 cells, we measured D=43 ± 8 µm² s⁻¹ in the cytosol. Taking this number and dividing it by the cube root of the molecular mass ratio between wild-type Clover-tagged β-catenin and Clover (120 kDa:26 kDa), we obtain an estimated D=26 ± 5 µm² s⁻¹ for the tagged β-catenin, which is on the high end of the distribution of D values of the respective ensemble (*Figure 6A*). Thus, Clover-tagged β-catenin may diffuse as an individual entity or as part of a smaller complex (compared to the one that mutant Cherry-tagged β-catenin is bound to).

From the autocorrelation amplitudes in the green and red color channels, the concentrations of wild-type Clover-tagged β-catenin and mutant Cherry-tagged β-catenin were determined as 32 ± 9 nM and 75 ± 11 nM (medians of distributions over more than 40 cells, *Figure 6A and B*). These data agree well with previous measurements in the RKO cell line, which revealed β-catenin concentrations of ~52 nM upon Wnt3A stimulation (*Hernández et al., 2012*) and in HEK293T and colon cancer cell lines (*Tan et al., 2012*).

Next, we were interested in whether activation of the Wnt signaling pathway had an impact on the diffusional dynamics of wild-type and mutant β-catenin. Therefore, FCS measurements were performed in the presence of the GSK3β inhibitor CHIR99021, which induces canonical Wnt signaling, as depicted in *Figure 4A*. Under control conditions (DMSO), wild-type β-catenin was less abundant and diffused faster than mutant β-catenin in the cytosol (*Figure 6C*). Interestingly, treatment with

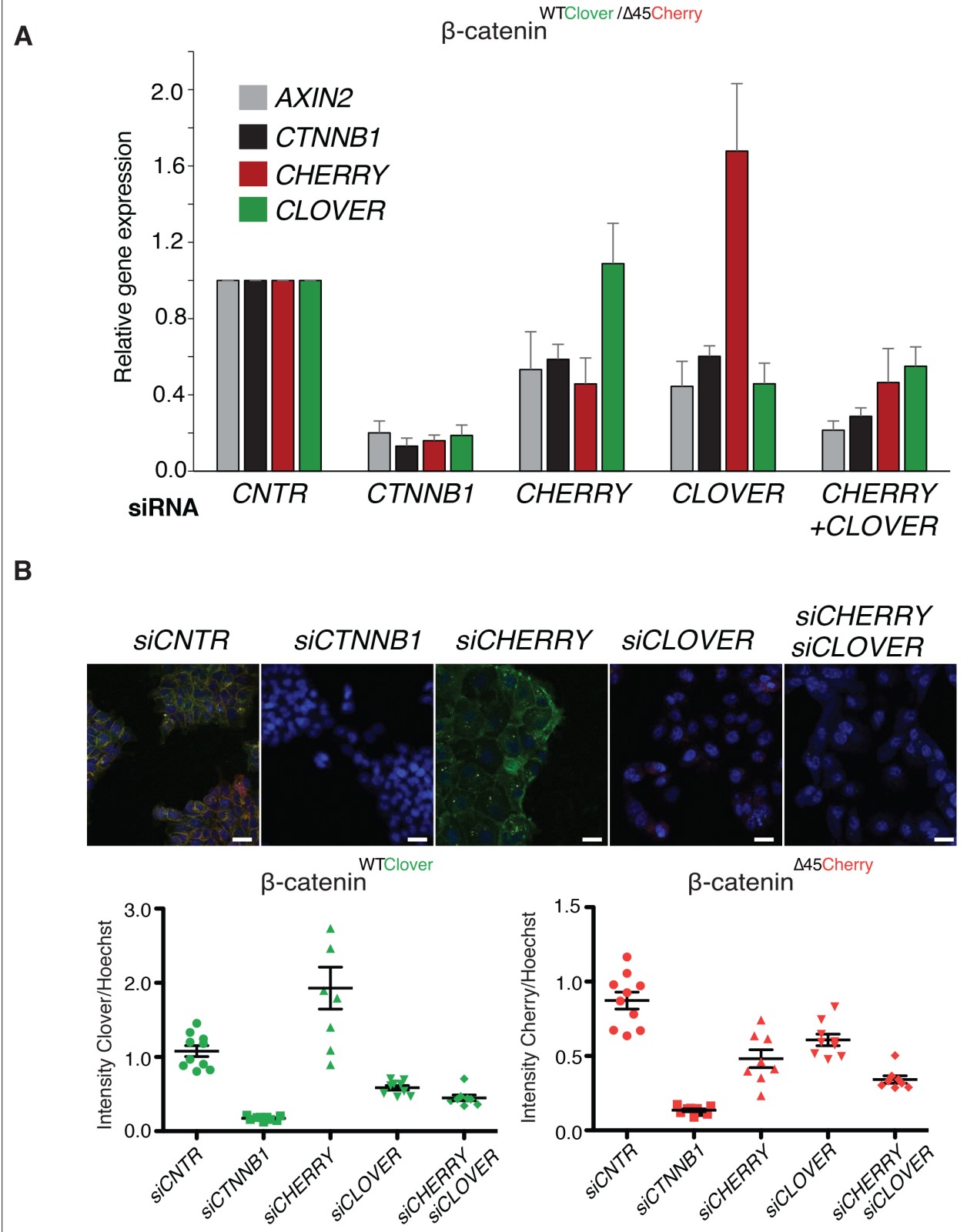

**Figure 5.** Wild-type and mutant β-catenin both contribute to Wnt pathway activation. (**A**) Expression levels of *AXIN2*, *CTNNB1*/β-catenin, *Cherry*, and *Clover* were measured 72 hr after knockdown with siRNAs directed against Clover, Cherry or both in HCT116 β-catenin[WTClover/Δ45Cherry] (n=4, mean ± SD). (**B**) Immunofluorescence analysis of HCT116 β-catenin[WTClover/Δ45Cherry] upon transfection with siRNAs targeting CTNNB1, CLOVER, CHERRY, or a combination of CLOVER and CHERRY (scale bar: 25 µm). Intensities of the Clover or Cherry signals were measured per slide and normalized to the

*Figure 5 continued on next page*

*Figure 5 continued*

intensity of Hoechst staining. 5–10 slides were measured per condition and are shown as dots. Representative one from four independent experiments (*Figure 5—figure supplement 1*) is shown.

The online version of this article includes the following source data and figure supplement(s) for figure 5:

**Source data 1.** Wild-type and mutant β-catenin both contribute to Wnt pathway activation.

**Figure supplement 1.** Immunofluorescence analysis of HCT116 β-catenin$^{WTClover/\Delta45Cherry}$ upon transfection with siRNAs targeting CTNNB1, CLOVER, CHERRY, or a combination of CLOVER and CHERRY.

CHIR99021 changed the diffusion coefficients and concentrations of wild-type β-catenin so that they became similar to the ones of mutant β-catenin, whereas smaller effects were observed for the mutant isoform. Essentially, the same effects were obtained for both isoforms in the nuclear fraction (*Figure 6C*). These observations suggest that, by applying the drug, wild-type β-catenin was also incorporated into a larger complex. The increased concentration suggests that complex formation reduces the probability of β-catenin degradation. Taken together, these results indicate that GSK3β inhibition renders the concentration and diffusional dynamics of wild-type β-catenin such that they are in the range of those of the mutant isoform.

The next question which we addressed was how inhibition of Wnt secretion influences concentration and dynamics of wild-type and mutant β-catenin. To this end, HCT116 β-catenin$^{WTClover/\Delta45Cherry}$ cells were treated with the porcupine inhibitor, LGK974, for 5–6 days before FCS measurements were performed (*Figure 6D*, *Figure 6—figure supplement 1B*). Upon this treatment, the concentration of wild-type β-catenin was found to be reduced, whereas the decrease of mutant β-catenin concentration was less pronounced. These concentration changes could be rescued by addition of recombinant Wnt3a, whereas addition of recombinant Wnt3a to cells without inhibitor treatment increased the wild-type β-catenin amount only slightly (*Figure 6D*, *Figure 6—figure supplement 1B*). Inhibition of Wnt secretion also affected the diffusional dynamics of Clover-tagged wild-type β-catenin (*Figure 6D*, *Figure 6—figure supplement 1B*). Upon LGK974 treatment, the diffusivity of the Clover-tagged β-catenin fraction was two times as large as for the control, $D_{WT,LGK}$=24.0 ± 2.8 μm² s$^{-1}$; however, the diffusivity of the mutant Cherry-tagged β-catenin fraction did not increase (*Figure 6D*, *Figure 6—figure supplement 1B*). Additional Wnt3a treatment reversed the effect of LGK974 treatment on the diffusivity of wild-type β-catenin. The FCS analysis yielded $D_{WT,LGK+Wnt3a}$=11.6 ± 2.3 μm² s$^{-1}$, which is essentially identical to that of the control.

## APC truncation affects the dynamics of the wild-type β-catenin isoform

*APC* mutations are frequently found in colorectal cancer and define the onset of the transition from adenoma to carcinoma. The complete loss of *APC* is very rare; truncating mutations lacking the C-terminal domain are, however, frequently observed (*Cancer Genome Atlas Network, 2012*; *Fearnhead et al., 2001*). Hence, we engineered cell lines carrying mutant truncated APC using CRISPR/Cas9 editing in the biallelically tagged HCT116 clone #37, and quantitatively assessed the impact of APC truncations on wild-type and mutant β-catenin protein in the same cell.

A sgRNA (sgAPCb) was designed by E-CRISP to target the mutation cluster region (MCR) in exon 15 of *APC* (*Heigwer et al., 2014*; *Zhan et al., 2019*), introducing a premature stop codon (*Figure 7A*). Edited single-cell clone was verified by amplicon sequencing confirming the homozygous *APC* mutation (*Figure 7A*). Subsequently, FCS measurements were performed to study the cytosolic and nuclear fractions of both parental HCT116 β-catenin$^{WTClover/\Delta45Cherry}$ and HCT116 β-catenin$^{WTClover/\Delta45Cherry}$ APC$^{LOF}$ (sgAPC) cells. In the cytosol (*Figure 7B*) and in the nucleus (*Figure 7—figure supplement 1A*) of APC$^{LOF}$ cells, the concentration and diffusional dynamics of wild-type β-catenin were different in comparison to cells with wild-type APC.

To investigate whether mutant β-catenin can still be found in the destruction complex, we performed immunoprecipitations with an APC antibody and with GFP/Clover and RFP/Cherry beads. Both wild-type β-catenin-Clover and mutant β-catenin-Cherry bind to APC, Axin1, and GSK3β (*Figure 7—figure supplement 1B*), indicating that both isoforms are part of the destruction complex. Both alleles of β-catenin can be regulated by the destruction complex but the level of regulation is different. Mutant β-catenin shows slight regulation upon inhibition of GSK3β or truncation of APC while the wild-type allele is more sensitive to these perturbations.

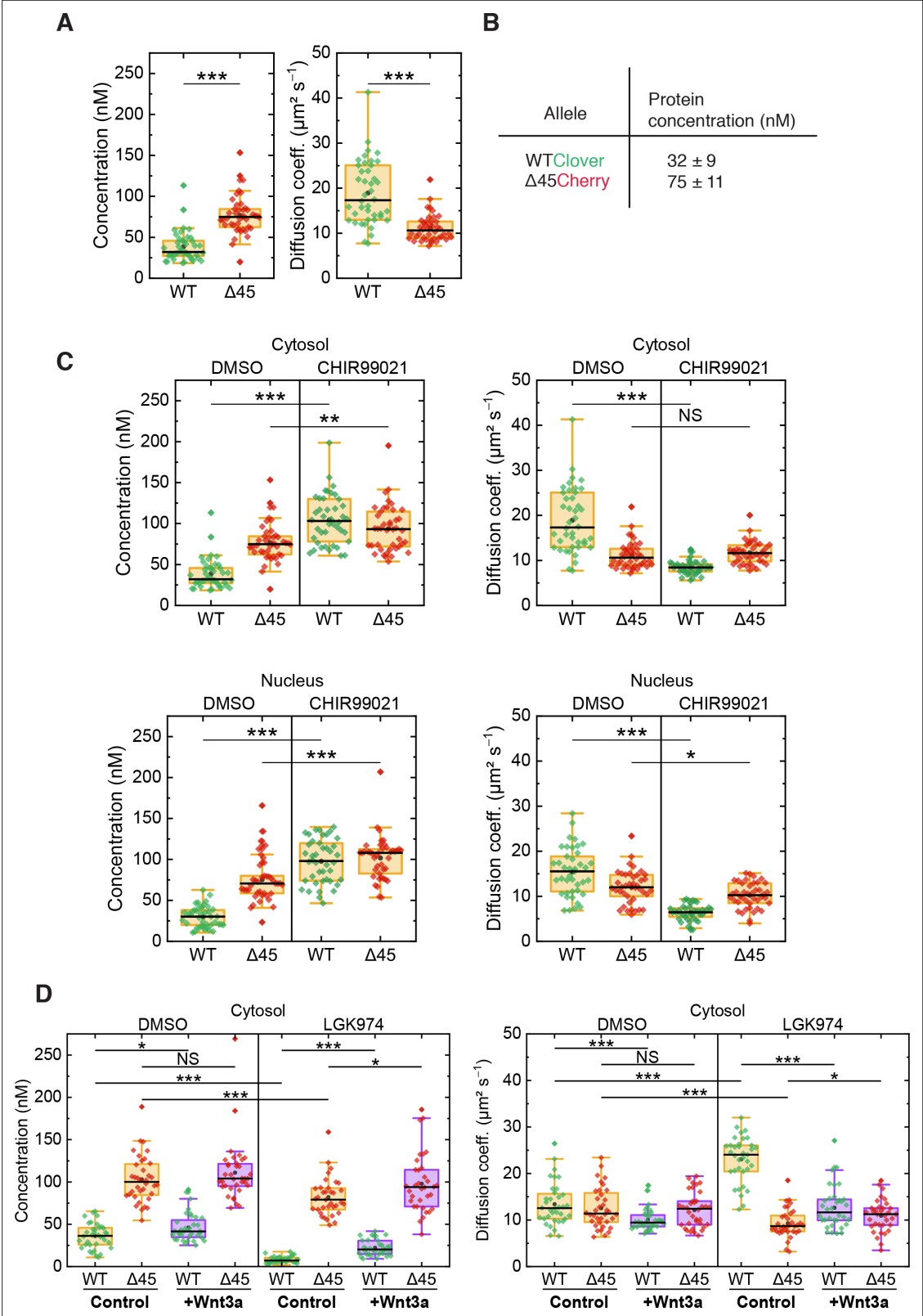

**Figure 6.** FCS autocorrelation analyses reveal differences in the dynamics and concentrations of wild-type and mutant β-catenin. (**A**) Concentrations (left) and diffusion coefficients (right) of Clover-tagged wild-type β-catenin and Cherry-tagged mutant β-catenin in the cytosol of HCT116 β-catenin[WTClover/Δ45Cherry]. (**B**) Protein concentrations (medians of the distributions shown in panel (**A**)). (**C**) Concentrations (left) and diffusion coefficients (right) of Clover-tagged wild-type β-catenin and Cherry-tagged mutant β-catenin measured on HCT116 β-catenin[WTClover/Δ45Cherry] cells that were treated for

*Figure 6 continued on next page*

*Figure 6 continued*

14–26 hr with either 10 µM CHIR99021 or DMSO as control. (**D**) HCT116 β-catenin^(wtClover/Δ45Cherry) cells were treated with 5 µM of LGK974 for 96 hr. Sixteen hours before termination of the experiment, 200 ng/ml of recombinant Wnt3a was added. FCS analysis was performed in the cytosol (data shown here) and in the nucleus (data shown in *Figure 6—figure supplement 1*). Each data point represents a 120 s FCS measurement in a single cell. In total, more than 40 cells per condition were measured in three independent experiments per box plot. p-values were calculated with the Mann-Whitney test (*** <0.001; ** <0.01; * <0.05; NS – non-significant). See also *Supplementary file 1*. FCS, fluorescence correlation spectroscopy.

The online version of this article includes the following source data and figure supplement(s) for figure 6:

**Source data 1.** FCS autocorrelation analyses reveal differences in the dynamics and concentrations of wild-type and mutant β-catenin.

**Figure supplement 1.** Cross-correlation analysis reveals that the two β-catenin protein isoforms diffuse independently.

## Discussion

How disease-causing genetic alleles affect protein function has remained a subject of intense research (*Sahni et al., 2015*). Mutations in β-catenin that activate the Wnt pathway occur in about 5% of colorectal and more than 25% of liver and uterine tumors (*Hoadley et al., 2018*; *Sanchez-Vega et al., 2018*). Nevertheless, the investigation of its biochemical properties has been hampered by the lack of suitable tools to study its function at a physiological level. Advances in genome editing methods such as CRISPR/Cas9 enable the modification of endogenous loci and the manipulations of genes for applications such as endogenous tagging of genes (*Cong et al., 2013*; *Dambournet et al., 2014*; *Mali et al., 2013*). Here, we generated genome engineered cell lines to simultaneously analyze wild-type and oncogenic mutant alleles of β-catenin. Using CRISPR/Cas9, we generated a bi-allelic endogenously tagged β-catenin cell model carrying Clover-tagged wild-type and Cherry-tagged mutant β-catenin. Recent reports described several approaches to increase the editing efficiency (*Lackner et al., 2015*; *Martin et al., 2019*; *Yang et al., 2014*). Based on these reports, we designed our two donor templates with 180 bp homology arms for the insert of around 3 kbp. Surprisingly, the efficiency of integration of the two donor templates was different, with Clover having about a 12-fold better integration efficiency than Cherry (*Figure 1—figure supplement 1B*). This might indicate that the efficiency of HDR depends not only on homology arms but also on the integrated sequence.

While tagging of proteins can interfere with protein functions, it can also lead to the accumulation of the protein and its potential overactivation. In line with our study, previous overexpression studies showed that C-terminal tagging of β-catenin did not affect its functions (*Giannini et al., 2000*; *Jamieson et al., 2011*; *Kafri et al., 2016*; *Krieghoff et al., 2006*; *Orsulic and Peifer, 1996*). We also show that all generated clonal cell lines with one or both tagged alleles had the same proliferation capacity (*Figure 3A*), indicating that the tag does not lead to a loss-of-function phenotype. In contrast, inactivation of β-catenin or canonical Wnt signaling in HCT116 cells has been previously shown to inhibit proliferation (*Scholer-Dahirel et al., 2011*; *Voloshanenko et al., 2013*). The region required for its adhesion function is within the central part of β-catenin and not at the C-terminus (*Valenta et al., 2012*; *Xing et al., 2008*). Therefore, the likelihood of C-terminal tagging affecting this function is rather small. Indeed, we found that β-catenin adhesive function was not affected (*Figure 3D and E*, *Figure 3—figure supplement 1B, C*). Previous studies used overexpression of fluorescently tagged β-catenin to analyze its kinetics (*Jamieson et al., 2011*; *Kafri et al., 2016*; *Krieghoff et al., 2006*). By contrast, here we studied the dynamics of endogenous wild-type and mutant β-catenin by leveraging genome engineering tools for endogenous tagging.

Using FCS, we analyzed mutant and wild-type proteins in the same cell. These measurements revealed that wild-type β-catenin diffuses faster than the mutant isoform in the cytoplasmic and the nuclear compartments, suggesting that wild-type protein is present as a monomer or as part of a smaller complex in the cell. In support of our data, a similar diffusion coefficient was recently determined by FCS for β-catenin endogenously tagged with monomeric SGFP2 in HAP1 cells (*de Man et al., 2021*). It appears that the mutant form of β-catenin binds more tightly to the destruction complex, resulting in slower diffusion in the cytoplasm. Consistent with this hypothesis, we found that the mutant protein physically interacts with APC, Axin, and GSK3β (*Figure 7—figure supplement 1*). Previous reports indicated that β-catenin mutants with deletion or mutation of Ser 45 retain signaling activity and are still responsive toward Wnt activation (*Chan et al., 2002*; *Parker et al., 2020*). We also found that mutant β-catenin is present at higher concentrations and diffuses more slowly than

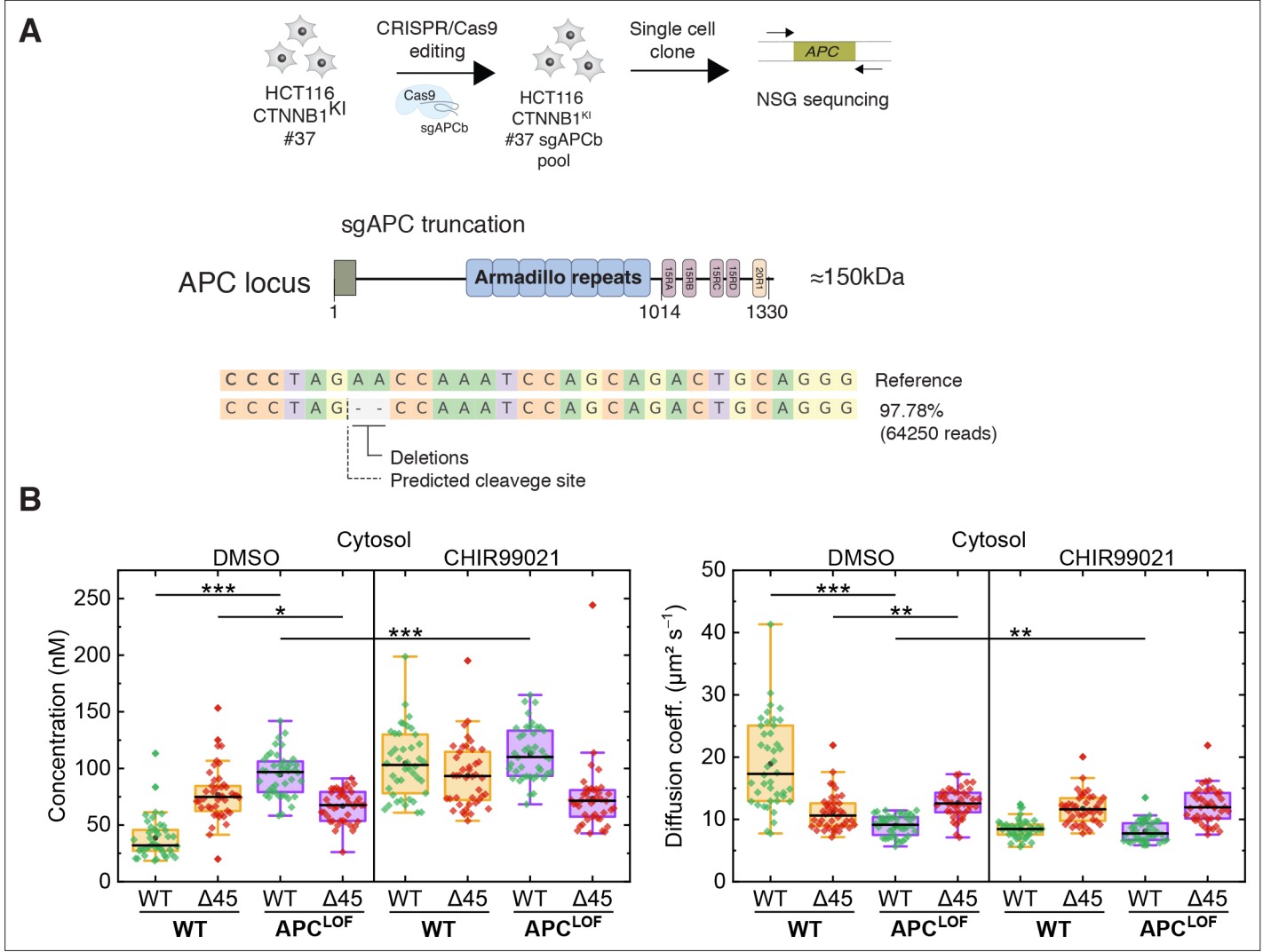

**Figure 7.** Truncation of APC affects abundance and diffusional dynamics of wild-type but not mutant β-catenin in the cytosol. (**A**) Schematic representation of the *APC* locus and target site of sgACP in the mutation cluster region (MCR) domain. (**B**) Concentrations (left) and diffusion coefficients (right) of Clover-tagged wild-type β-catenin and Cherry-tagged mutant β-catenin in the cytosol of HCT116 β-catenin^WTClover/Δ45Cherry and sgAPC targeted clone (APC^LOF) cells that were treated for ~16 hr with either 10 μM CHIR9901 or DMSO as control. Subsequently, FCS measurements were performed in the cytosol (data shown here) and in the nucleus (data shown in *Figure 7—figure supplement 1A*). Each data point in the box plots represents a result from a 120 s FCS measurement in a single cell. Per box plot, more than 40 cells were investigated in three independent experiments. p-values were calculated with the Mann-Whitney test (*** <0.001; ** <0.01; * <0.05; NS – non-significant). The exact values are provided in the *Supplementary file 1*. FCS, fluorescence correlation spectroscopy.

The online version of this article includes the following source data and figure supplement(s) for figure 7:

**Source data 1.** Truncation of APC affects abundance and diffusional dynamics of wild-type but not mutant β-catenin in the cytosol.

**Source data 2.** Truncation of APC affects abundance and diffusion of wild-type but not mutant β-catenin in the nucleus.

**Figure supplement 1.** Truncation of APC affects abundance and diffusion of wild-type but not mutant β-catenin in the nucleus.

wild-type protein in the nucleus, suggesting increased nuclear import and binding to other proteins and DNA to form larger complexes.

Activation of Wnt signaling by inhibition of GSK3β or truncation of APC results in slower diffusion and higher abundance of wild-type β-catenin, indicating that β-catenin now becomes part of larger multi-protein complexes. This might—at least in part—be due to enhanced shuttling into the nucleus, where β-catenin interacts with the Wnt enhanceosome, a large transcriptional complex including TCF/LEF, PYGO, BCL9, and other proteins to activate transcription of Wnt target genes (*Fiedler et al., 2015*; *van Tienen et al., 2017*). Along this line, FRAP analysis demonstrated that TCF4, APC, Axin,

and Axin2 reduced β-catenin mobility and nucleo-cytoplasmic shuttling suggesting that these inter-action partners control the subcellular localization of β-catenin by sequestering it in the respective compartment (*Krieghoff et al., 2006*). Accordingly, *Goentoro and Kirschner, 2009* demonstrated that structural changes of β-catenin rather than molecule numbers regulate Wnt signaling (*Goentoro and Kirschner, 2009*). Furthermore, a recent FCS-based study in HAP1 cells showed that Wnt pathway activation regulates the distribution of free β-catenin and β-catenin bound in multi-protein complexes between nucleus and cytosol by modulating β-catenin destruction, shuttling between cytoplasm and nucleus, and nuclear retention (*de Man et al., 2021*).

In the present work, we have used FCS to quantify the concentration and diffusional mobility of β-catenin in the nuclear and cytosolic compartments for the first time. The observed small diffusion coefficients indicate that β-catenin migrates as part of multi-protein complexes. Unraveling the protein composition of these complexes will require dual- or multicolor FCS experiments, which allow us to analyze co-diffusion of proteins via cross-correlation analysis (*Eckert et al., 2020*). Further-more, we have performed FCS experiments at single locations within subcellular compartments. In the future, we will also adopt fluorescence microscopy-based fluctuation spectroscopy techniques for the analysis of intracellular protein dynamics in a spatially resolved fashion (*Tetin, 2013*). While these tech-niques are frequently used with conventional, diffraction-limited microscopy, super-resolution variants have also become available (*Hedde et al., 2013*; *Gao et al., 2017*).

β-catenin plays an important role not only in Wnt signaling but also in cell adhesion, in which it links the adhesion complex to the actin cytoskeleton via its interaction with the actin-binding protein α-catenin (*Rimm et al., 1995*). Since the different functions require distinct localization, β-catenin can be found in at least two distinct pools, one at the cell membrane and one that can shuttle between the cytosol and nucleus. Whether both pools originate from a common pool or from two independent pools, and how these reservoirs regulate each other is still a matter of debate. Accordingly, it was shown that it is possible to in vivo affect one function of β-catenin without interfering with the other (*Orsulic and Peifer, 1996*; *Valenta et al., 2012*; *Valenta et al., 2011*). Moreover, different molecular forms of β-catenin have been described, demonstrating that most cytosolic β-catenin exists in a mono-meric form, whereas dimers of α-catenin and β-catenin interacted mainly with cadherin (*Gottardi and Gumbiner, 2004*). Interestingly, our immunoprecipitation and FCS analyses indicate that wild-type and mutant β-catenin are found in two separate pools. We observed that β-catenin is mobile in both compartments, with mutant β-catenin being found in higher molecular weight complexes.

In summary, we engineered, to our knowledge, the first biallelically tagged cell model carrying a different fluorescent fusion protein on each allele, demonstrating the feasibility of our one-step tagging strategy to analyze the consequences of genetic variants. β-catenin knock-in cell lines provide a powerful tool for investigating β-catenin's role in the canonical Wnt signaling, and also in adhesion. Due to these properties of β-catenin, these cell models can serve to evaluate both functions in parallel on the endogenous level. Cell models of wild-type and mutant variants can also be used for drug discovery, identifying small molecules that only interfere with mutant, but not wild-type β-catenin, thereby circumventing side effects that are often associated with interfering with homeostatic levels of Wnt signaling. In general, bi-allelic tagging of genetic variants in their endogenous locus can open new avenues toward understanding and possibly targeting disease-causing variants in a broad spec-trum of cellular and organismal phenotypes.

# Materials and methods

**Key resources table**

| Reagent type (species) or resource | Designation | Source or reference | Identifiers | Additional information |
|---|---|---|---|---|
| Cell line (*Homo sapiens*) | HCT116 | ATCC | CCL-247/CVCL_0291 | Human colon cancer cells |
| Strain, strain background (*Escherichia coli*) | Top 10 *E. coli* | Life Technologies GmbH | C404003 | Chemically competent cells |
| Other | TransIT-LT1 | VWR | 731-0029 | Plasmid transfection reagent |

*Continued on next page*

*Continued*

| Reagent type (species) or resource | Designation | Source or reference | Identifiers | Additional information |
|---|---|---|---|---|
| Other | Lipofectamine RNAiMax | Life technologies GmbH | 13778150 | siRNA transfection reagent |
| Transfected construct (human) | siRNA:UBC | GE Healthcare Dharmacon/Horizon | MU-019408-01-0002 | #1 GTGAAGACCCTGACTGGTA<br>#2 AAGCAAAGATCCAGGACAA<br>#3 GAAGATGGACGCACCCTGT<br>#4 GTAAGACCATCACTCTCGA |
| Transfected construct (human) | siRNA:Non targeting | GE Healthcare Dharmacon/Horizon | D-001810-02 | UGGUUUACAUGUUGUGUGA |
| Transfected construct (human) | siRNA:Control | Ambion | S29712 | |
| Transfected construct (human) | siRNA:GFP | GE Healthcare Dharmacon//Horizon | D-001300-01-05 | GCAAGCTGACCCTGAAGTTC |
| Transfected construct (human) | siRNA:CTNNB1 | Ambion | S438 | CUGUUGGAUUGAUUCGAAAtt |
| Transfected construct (human) | siRNA:Cherry | IDT | Custom design | rCrArU rGrGrC rCrArU rCrArU rCrArA rGrGrA rGrUrU rCrArU rG |
| Antibody | Anti-β-actin HRP, (Rabbit polyclonal) | Santa Cruz Biotechnology | Cat#: 47778/ RRID:AB_2714189 | WB (1:20,000) |
| Antibody | Anti-β-actin, (Rabbit polyclonal) | Santa Cruz Biotechnology | Cat#: 47778/ RRID:AB_2714189 | WB (1:40,000) |
| Antibody | Anti-β-catenin, (Mouse monoclonal) | Dianova/Affinity BioReagent | Cat#: MA1-2001/ RRID:AB_326078 | WB (1:3000), IF (1:500) |
| Antibody | Anti-Cherry, (Mouse monoclonal) | ClonTech | Cat#: 632543/ RRID:AB_2307319 | WB (1:1000) |
| Antibody | Anti-GFP, (Mouse monoclonal) | Invitrogen | Cat#: 332600/ RRID:AB_2533111 | WB (1:1000) |
| Antibody | Anti-GFP, (Rabbit polyclonal) | Invitrogen | Cat#: A6455/ RRID:AB_221570 | WB (1:1000) |
| Antibody | Anti-E-Cadherin, (Mouse monoclonal) | BD Biosciences | Cat#: 610182/ RRID:AB_397581 | WB (1:1000) |
| Antibody | Anti-V5, (Rabbit polyclonal) | Rockland | Cat#: 600-401−378/ RRID:AB_828437 | WB (1:2000) |
| Antibody | Anti-V5, (Mouse monoclonal) | Thermo Fisher Scientific | Cat#: 15253/ RRID:AB_10977225 | WB (1:1000) |
| Antibody | Anti-Flag, (Rabbit polyclonal) | Sigma-Aldrich | Cat#: F7425/ RRID:AB_439687 | WB (1:1000) |
| Antibody | Anti-Flag, mouse (Mouse monoclonal) | Sigma-Aldrich | Cat#: F3165/ RRID:AB_259529 | WB (1:1000) |
| Antibody | Anti-APC ALI 12–28, (Mouse monoclonal) | Santa Cruz Biotechnology | Cat#: sc-53165/ RRID:AB_628734 | WB (1:1000) |
| Antibody | Anti-Axin1 C76H11, (Rabbit polyclonal) | Santa Cruz Biotechnology | Cat#: 2087/ RRID:AB_2274550 | WB (1:1000) |
| Antibody | Anti-GSK3β D5C5Z, (Rabbit polyclonal) | Cell Signaling Technology | Cat#: 12456/ RRID:AB_2636978 | WB (1:1000) |
| Antibody | Normal IgG, (Rabbit) | Cell Signaling Technology | Cat#: 2729/ RRID:AB_1031062 | WB (1:1000) |
| Antibody | IgG1 K isotype control (Mouse) | eBioscience | Cat#: 16-4714-81/ RRID:AB_470160 | IP (1:1000) |

*Continued on next page*

*Continued*

| Reagent type (species) or resource | Designation | Source or reference | Identifiers | Additional information |
|---|---|---|---|---|
| Antibody | Anti-mouse IgG-HRP (Goat) | Jackson ImmunoResearch | Cat#: 115-035-003/ RRID:AB_10015289 | WB (1:10,000) |
| Antibody | Anti-rabbit IgG-HRP (Goat) | Jackson ImmunoResearch | Cat#: 111-035-003/ RRID:AB_2313567 | WB (1:10,000) |
| Antibody | True Blot ULTRA Anti-mouse IgG-HRP | eBioscience | Cat#: 18-8817-33/ RRID:AB_2610851 | WB (1:5000) |
| Recombinant DNA reagent | pRL actin-Renilla | *Nickles et al., 2012* | | Renilla luciferase reporter |
| Recombinant DNA reagent | pgl4.23 TCF4/Wnt-luciferase | *Demir et al., 2013* | | TCF4/Wnt-Firefly Luciferase reporter |
| Recombinant DNA reagent | px459 | *Mali et al., 2013* | RRID:SCR_002037 | Cloning of the sgRNA |
| Recombinant DNA reagent | px459sgCTNNB1 | This paper | | See Materials and methods, *Figure 1—figure supplement 1*– sgRNA: TGACCTGTAAATCATCCTTT |
| Recombinant DNA reagent | px459sgAPC#b | This study | | See Materials and methods, *Figure 7A* sgRNA: TAGAACCAAATCCAGCAGA |
| Recombinant DNA reagent | pSpCas9(BB)–2A-GFP (PX458) | *Mali et al., 2013* | RRID:SCR_002037 | Control vector |
| Recombinant DNA reagent | pMK-RQ HA-FLAG-mClover-PGK-HygRHA | This study | | Donor template, See Materials and methods, *Figure 1—figure supplement 1* |
| Recombinant DNA reagent | pMK-RQ HA-V5-mCherry-PGK-BRS-HA | This study | | Donor template, See Materials and methods, *Figure 1—figure supplement 1* |
| Chemical compound, drug | CHIR99021 | Merck Millipore | 361571 | GSK-3β inhibitor |
| Chemical compound, drug | LGK974 | Hölzel Diagnostika | TRC-L397640-50mg | Porcupine inhibitor |
| Peptide, recombinant protein | Mouse Wnt3a | PeproTech | 315-20-10 | |
| Other | Triton X-100 | Sigma-Aldrich | T8787-250ml | |
| Other | NP-40 | Sigma-Aldrich | NP40S-100ml | |
| Other | GFP-(gta) magnetic beads/agarose | Chromotec | gtak-20 gtma-20 | |
| Other | RFP-Trap(rta) magnetic beads/agarose | Chromotec | rta-20 rtma-20 | |
| Commercial assay or kit | BCA Protein Assay Kit | Thermo Fisher Scientific | 23225 | |
| Other | 4–12% NuPAGE Bis-Tris gels | Thermo Fisher Scientific | NW04122BOX; NW00120BOX | |
| Other | 3–8% NuPAGE Tris acetate gels | Thermo Fisher Scientific | EA03752BOX | |
| Other | Nitrocellulose membranes | GE Healthcare | GE10600002 | |
| Commercial assay or kit | In-Fusion HD Cloning | Takara | 639650 | |
| Other | Dynabeads Protein G magnetic beads | Thermo Fisher Scientific | 10004D | |

*Continued on next page*

*Continued*

| Reagent type (species) or resource | Designation | Source or reference | Identifiers | Additional information |
|---|---|---|---|---|
| Other | ECL reagent | Merck Millipore | WBKLS0100 | |
| Other | ECL reagent | BiozolDiagnostica | MBL-JM-K820-500 | |
| Other | Hyperfilm ECL; 18×24 cm² | Amersham/GE Healthcare | GE28-9068-36 | |
| Other | Puromycin | Sigma-Aldrich | P9620 | |
| Other | Hygromycin | Gibco/Thermo Fisher Scientific | 10687010 | |
| Other | Blasticidin | Life Technologies GmbH | R21001 | |
| Commercial assay or kit | DNeasy Blood & Tissue Kit | QIAGEN | 69504 | |
| Commercial assay or kit | RevertAid H Minus First Strand cDNA Synthesis Kit | Thermo Fisher Scientific/VWR | K1632 | |
| Commercial assay or kit | QIAfilter Plasmid Maxi Kit | QIAGEN | 12263 | |
| Commercial assay or kit | Qiagen RNeasy Mini Kit | QIAGEN | 12571 | |
| Other | McCoy | Life Technologies GmbH | 26600080 | |
| Commercial assay or kit | Light Cycler 480 Probes Master Mix QPCR | Roche | 4887301001 | |
| Other | Q5 Hot Start High-Fidelity DNA Polymerase | New England Biolabs | M0493S | |
| Other | dNTP Set 100 mM | VWR/Fermentas/Thermo Fisher Scientific | R0182 | |
| Commercial assay or kit | Light Cycler 480 Probes Master Mix QPCR | Roche | 4887301001 | |
| Other | PFA/ paraformaldehyde | VWR | 43,368.9 L | |
| Other | 4% paraformaldehyde in PBS | Santa Cruz Biotechnology | sc-281692 | |
| Other | Vectashield DAPI solution | Biozol Diagnostica | C-H-1200 | |
| Other | Hoechst 33342; trihydrochloride; trihydrate | Life Technologies GmbH | H1399 | |
| Other | BSA | Gerbu | 5010500 | |
| Other | PBS | Sigma-Aldrich | P3813-10PAK | |
| Other | Goat serum | Cell Signaling Technology | 5425S | |
| Other | Microscope slides 76×26 mm² | Carl Roth GmbH | H868.1 | |
| Other | µ-Slide eight well | Ibidi | 80826 | |
| Sequence-based reagent | Universal probe library #011 | Roche/Sigma-Aldrich | 4685105001 | |
| Sequence-based reagent | Universal probe library #148 | Roche/Sigma-Aldrich | 04685148001 | |
| Sequence-based reagent | Universal probe library #152 | Roche/Sigma-Aldrich | 4694384001 | |

*Continued on next page*

*Continued*

| Reagent type (species) or resource | Designation | Source or reference | Identifiers | Additional information |
|---|---|---|---|---|
| Sequence-based reagent | Universal probe library #088 | Roche/Sigma-Aldrich | 4689135001 | |
| Sequence-based reagent | Universal probe library #060 | Roche/Sigma-Aldrich | 4688589001 | |
| Sequence-based reagent | Universal probe library #021 | Roche/Sigma-Aldrich | 4686942001 | |
| Sequence-based reagent | sgRNA used for targeting CTNNB1 (px459sgCTNNB1) | This paper | | TGACCTGTAAATCATCCTTT |
| Sequence-based reagent | sgRNA used for targeting APC (px459sgAPC#b) | This paper | | TAGAACCAAATCCAGCAGA |
| Software, algorithm | Adobe Photoshop CS6 | Adobe | RRID:SCR_014199 | |
| Software, algorithm | Adobe Illustrator CS6 | Adobe | RRID:SCR_010279 | |
| Software, algorithm | Adobe Affinity Designer | Adobe | RRID:SCR_016952 | |
| Software, algorithm | Fiji | | PRID:SCR_002285 | |
| Software, algorithm | ImageJ | | RRID:SCR_003070 | |
| Software, algorithm | Biorender | | RRID:SCR_018361 | |
| Software, algorithm | OriginPro | | RRID:SCR_014212 | |
| Software, algorithm | MATLAB | | RRID:SCR_013499 | |
| Other | Fibronectin | Sigma-Aldrich | F1141-5MG | |
| Other | DPBS | Gibco (ThermoFisher Scientific) | 14190-144 | |
| Other | 8-well Nunc Lab-Tek chambered cover glass | Thermo Fisher Scientific | 155411 (#1) | |
| Other | McCoy's 5A - w/ L-Gln, w/o Phenol Red McCoy's 5AMedium w/ L-Glutamine w/o Phenol red and Sodium bicarbonate | GE Lifesciences/HyClone (Thermo Fisher Scientific) HIMEDIA (NeoLab) | SH30270.01/10358633 AT179-5L | Sodiumbicar-bonate was added before sterile filtration |
| Other | Alexa 488 | Thermo Fisher Scientific | 10266262 | Reference dye |
| Other | Alexa 546 | Thermo Fisher Scientific | 10534783 | Reference dye |
| Other | Xfect | Takara ClonTech | 631318 | Transfection reagent |
| Sequence-based reagents | Primers (*Supplementary files 5 and 6*) | Eurofins | | See *Supplementary files 5 and 6* |

## Cell Culture

The parental colon cancer HCT116 (β-catenin$^{WT/\Delta45}$) cell line was purchased from the American Type Culture Collection. HCT116 β-catenin$^{WT+/-}$ cells were obtained from Horizon Discovery. Cells were culture in McCoy's medium (Gibco) supplemented with 10% fetal bovine serum (Biochrom GmbH). Cells were grown at 37°C and 10% $CO_2$ in a humidified atmosphere. Cells were tested for cross-contamination using SNP profiling by a Multiplex human Cell Authentication (MCA) assay (Multiplexion) and regularly checked for Mycoplasma contamination.

## Transfection

Transient transfections were performed using TransIT-LT1 Transfection Reagent (731-0029; VWR) for DNA plasmids and Lipofectamine RNAiMax (Thermo Fisher Scientific) for siRNAs according to the manufacturer's protocol. 15–25 nM Dharmacon siRNAs (Horizon Discovery) and 5–10 nM Ambion siRNAs (Thermo Fisher Scientific) were used for reverse siRNA transfection (*Supplementary file 2*).

## Small-molecule inhibitors

GSK-3β inhibitor CHIR99021 (Cat. no. 361571) was obtained from Merck Millipore. ICG001 (Cat. no. Cay16257-1) was obtained from Biomol. DMSO was used as the vehicle control. LGK974 (TRC-L397640-50mg) was synthesized by Hölzel Diagnostika.

## Western blot and co-immunoprecipitations assays

Whole-cell lysates were extracted using a buffer containing non-ionic detergents such as Triton X-100 or NP-40, supplemented with protease inhibitors (Roche). Protein concentration was determined using BCA Protein Assay Kit (Thermo Fisher Scientific) according to the manufacturer's instructions. Samples were loaded on 4–12% NuPAGE Bis-Tris gels or 3–8% NuPAGE Tris acetate gels (Thermo Fisher Scientific) and transferred to the nitrocellulose membranes (GE Healthcare, GE10600002) following standard Western blotting procedure. Antibodies and their dilutions are listed in *Supplementary file 3*. Clover- and Cherry-tagged proteins were immunoprecipitated by incubation with either GFP-(gta) or RFP-Trap(rta) magnetic beads/agarose (Chromotek) for 1 hr at 4°C with rotation. For E-cadherin and β-catenin IPs extracts were incubated with control or respective antibody (*Supplementary file 3*) together with Dynabeads Protein G magnetic beads (Thermo Fisher Scientific) for 14–16 hr at 4°C. For signal visualization, blots were incubated with ECL reagent (WBKLS0100; Merck Millipore). Full Western blot scans are provided in the Source Data files. Adobe Photoshop was used for equal adjustment of brightness and contrast across the whole image.

sgRNA design and cloning sgRNAs sequences targeting the *CTNNB1* and *APC* genes were designed using E-CRISP (*Heigwer et al., 2014*), synthesized by Eurofins and cloned into the px459 plasmid (#62988, Addgene) (*Supplementary file 4*), according to a previously described protocol (*Mali et al., 2013*). To generate donor plasmids the desired homology arm's regions and epitope tag (either V5 or FLAG) were synthetized by Invitrogen GeneArt Gene Synthesis and cloned into the vector backbone pMK-RQ. The Clover PGK-HygR and Cherry PGK-BRS were cloned from the published vectors pMK278 (Addgene #72794) and pMK282 (Addgene #72798), respectively (*Natsume et al., 2016*). PCR primers are listed in *Supplementary file 6*.

## Generation and validation of allele-specific endogenously tagged β-catenin cell lines

HCT116 cells were transfected with px459sgCTNNB1 and the two donor template-encoding plasmids KOZAK_FLAG_mClover3 and KOZAK_V5_mCherry2 using TransIT (731-0029; VWR) according to manufacturer's instructions. As controls, each donor template was also transfected alone together with px459sgCTNNB1 to generate single-tagged β-catenin cells and to adjust gates for FACS sorting. After 72 hr, cells were selected with 2 µg/ml puromycin (P9620, Sigma-Aldrich). After 48 h, 100 µg/ml hygromycin 10687010 (Gibco/Thermo Fisher Scientific) and 10 µg/ml blasticidin (R21001, Life Technologies GmbH) were added to select edited cells for 5 days. Subsequently, surviving cells were sorted as single-cell clones by FACS into 96-well plates for cultivating. To validate correctly endogenously tagged β-catenin clones genotyping was performed. Genomic DNA was extracted with phenol-chloroform or with DNeasy Blood & Tissue Kit (QIAGEN, 69504). Different primer pair combinations were designed for genotyping (*Supplementary file 6*). Primer pair (a) binds to the endogenous β-catenin locus flanking the donor cassette indicating homo- or heterozygous integration of the donor template. Primer pairs (b), (b'), and (c) bind outside and inside the donor indicating the correct in-frame integration. Primer pair (d) binds inside the donor indicating which fluorophore was integrated. PCR products were analyzed by agarose gel electrophoresis.

## TCF4/Wnt-reporter activity assay

To determine Wnt signaling activity, the luciferase-based dual Wnt reporter assay was performed as described previously (*Demir et al., 2013*). Briefly, cells were transfected with plasmids encoding the

TCF4/Wnt-driven firefly luciferase and actin-promoter driven Renilla luciferase as control (*Supplementary file 4*). Dual-luciferase readout was performed 48 hr after transfection using Mithras LB940 plate reader (Berthold Technologies). Wnt activity was calculated by normalization of the TCF4/Wnt-luciferase values to the actin-Renilla values.

## RT-qPCR

Total RNA from cell pellets was isolated with the RNeasy Mini Kit (QIAGEN) according to the manufacturer's protocol. RNA was reverse-transcribed into cDNA using the RevertAid H Minus First Strand cDNA Synthesis Kit (Thermo Fisher Scientific). qPCR was performed using the Universal probe library (Roche) with the LightCycler 480 (Roche) instrument according to the manufacturer's instructions. A list of primers used in this paper for qPCR is shown in *Supplementary file 5*.

## Immunofluorescence assays

Sterile coverslips (Thermo Fisher Scientific) were placed into 12-well plates and cell suspension was added. After 1 day, cells were treated with the indicated drugs for 20–24 hr. Cells were fixed using 4% PFA (VWR) for 20 min at room temperature. For antibody staining, cells were permeabilized with 0.25% Triton/phosphate-buffered saline (PBS). After several washing steps, cells were blocked for 1 hr with 5% goat serum (v/v) and 5% BSA (v/v) in PBS at room temperature. After overnight incubation with the primary antibody at 4°C, cells were incubated with fluorescently labeled secondary antibodies (1:250) for 1 hr at room temperature in the dark. Antibodies were diluted in PBS with 5% goat serum (v/v) and 5% BSA (v/v). Finally, cover slips were gently transferred onto microscope slides (Carl Roth GmbH) and fixed with Vectashield mounting media containing DAPI solution (Biozol Diagnostica). Fluorescence images were acquired with a Leica TCS SP5 confocal microscope. For live-cell imaging, cells were added into μ-slide with a glass bottom (Ibidi) and staining was performed as described. For automated microscopy, cells were seeded into 384-well plates and subjected to fixation using the CybiWell Vario robotic system. Fluorescence images were acquired using InCell Analyzer 2200 microscope (GE Healthcare) at 20× magnification in three channels (DAPI, Cy3, and FITC) with four sites per well.

For LGK974 treatment, cells were seeded into wells of a six-well plate and treated with Control/DMSO or 5 μM LGK974 for 48/72 hr, then washed several times with PBS and split into the Ibidi μ-Slide (80826) and wells of a six-well plate. After an additional treatment with LGK974/Ctrl for 32 hr, cells were stimulated with 200 ng/ml of recombinant Wnt3a for 16 hr. Cells were fixed with 4% PFA for 10 min and stained with the Hoechst solution (1:2000 in PBS) for 30 min. Fluorescence images were acquired with a Leica TCS SP8 confocal microscope. 5–20 slides per condition were taken. The fluorescence intensities of Clover, Cherry, or Hoechst were analyzed with ImageJ without any additional manipulation of the images. The Clover and Cherry intensities were normalized to Hoechst and further to control.

In the silencing experiment, CTNNB1, CLOVER, and CHERRY were downregulated by applying siRNA directly on the Ibidi μ-Slide for 72 hr. Subsequently, the staining and analysis were performed similar to the LGK974 experiment.

## Image analysis

Image analysis of automatic microscopy was performed using R package EBImage (*Pau et al., 2010*) and adapted based on previous analysis methods (*Carpenter et al., 2006*; *Fuchs et al., 2010*). Nuclei and cell body were segmented and features for intensity, shape, and texture were extracted for each cell based on the DAPI, actin, and tubulin staining. Features were then summarized per experiment by mean calculation over all cells. As a proxy for cell count, the number of segmented nuclei was used. Images were assembled using Affinity Designer, Biorender, and Adobe Illustrator.

## Fluorescence correlation spectroscopy

To prepare fibronectin-coated sample chambers, fibronectin was diluted to 50 μg ml$^{-1}$ in Dulbecco's phosphate buffered saline (DPBS, no calcium, no magnesium, Sigma-Aldrich). Each well of a eight-well Nunc Lab-Tek chambered cover glass (Thermo Fisher Scientific) was incubated with 200 μl of the solution for at least 1 hr at room temperature. Afterward, the solution was removed by aspiration, and the wells were allowed to dry. These chambers were either used immediately or stored at 4°C for a

maximum of 1–2 weeks and washed with DPBS directly before use. For FCS measurements, HCT116 cells were cultured in McCoy's medium without phenol red in the chambers. For drug treatment, the cell culture medium was replaced with a medium containing 10 µM CHIR99021 ~16–26 hr before the FCS measurements. For LGK974 treatment, cells were seeded in six-well plates and incubated with 5 µM LGK974 for 5 days. During this period, the drug-containing medium was renewed and, if necessary, cells were split. On day 6, cells were transferred into fibronectin-coated eight-well chambered cover glass, where they were left to adhere for ~4–6 hr before adding 100 ng ml$^{-1}$ of recombinant Wnt3a. FCS measurements were started at ~16–26 hr after Wnt treatment. All FCS measurements were carried out at 37°C and 5% $CO_2$.

For FCS analysis, intensity time traces were acquired in two color channels for 120 s each by using a home-built confocal microscope described previously (*Eckert et al., 2020*). mClover3 and mCherry2 were excited with pulsed 470 nm and 561 nm laser light of typically 3 µW (1.5 kW cm$^{-2}$) and 5 µW (2.8 kW cm$^{-2}$), respectively. The emission was filtered by 525/50 nm and 607/37 nm (center/width) band-pass filters (BrightLine, Semrock, Rochester, NY). For data acquisition, the observation volume of the microscope was first positioned into the cytoplasm and directly afterward into the nucleus. The intensity decrease during the measurement due to photobleaching was compensated as described (*Dörlich et al., 2015*). Then, autocorrelation and cross-correlation functions were calculated from the time traces, and correlation amplitudes and times were extracted by fitting a model function to the correlation functions that describes free diffusion of fluorescent particles through an observation volume shaped as a three-dimensional Gaussian function (*Gao et al., 2017*). The correlation amplitudes of each individual cell measurement were corrected for photobleaching according to *Eckert et al., 2020*, and uncorrelated background according to *Foo et al., 2012*. For the latter correction, we measured background intensities in the cytosol and in the nucleus on ensembles of more than 15 cells of lines #45 and #33, both under control conditions (DMSO) and with drug treatment (CHIR99021). For each cell measurement, the correction factor was calculated using the median intensity of the respective ensemble. Finally, the (microscope-dependent) correlation times and corrected amplitudes were converted into (microscope-independent) diffusion coefficients and concentrations, which are reported in this work. For this conversion, we carried out careful FCS calibration experiments using solutions of organic dyes (Alexa 488 and Alexa 546, Thermo Fisher Scientific) with well-known diffusion coefficients and concentrations. Furthermore, FCS measurements of fluorescent proteins (eGFP, Clover, and Cherry) overexpressed in HCT116 wild-type cells were performed to obtain estimates of the molecular mass of the β-catenin carrying entities.

## Acknowledgements

The authors are grateful to F Port, KE Boonekamp, and S Redhai for providing critical comments on the manuscript. The authors would like to acknowledge all members of the Boutros group and of the SFB1324 for helpful discussions. The authors thank F Zhang, M Waterman, and D Virshup for plasmids, which were obtained via Addgene. The authors are grateful for the support by the Excellence Cluster CellNetworks Core Facilities. Research in the labs of MB and GUN has been supported by the SFB1324 "Mechanisms and Functions of Wnt Signaling" of the German Research Foundation (DFG).

## Additional information

### Funding

| Funder | Grant reference number | Author |
|---|---|---|
| Deutsche Forschungsgemeinschaft | SFB1324, Project number 331351713 | Giulia Ambrosi Oksana Voloshanenko Antonia Franziska Eckert Dominique Kranz Gerd Ulrich Nienhaus Michael Boutros |
| Helmholtz Association | Material Systems Engineering | G Ulrich Nienhaus |

| Funder | Grant reference number | Author |
|--------|------------------------|--------|

The funders had no role in study design, data collection and interpretation, or the decision to submit the work for publication.

## Author contributions

Giulia Ambrosi, Conceptualization, Investigation, Methodology, Visualization, Writing – original draft; Oksana Voloshanenko, Conceptualization, Investigation, Methodology, Validation, Visualization, Writing – original draft, Writing – review and editing; Antonia F Eckert, Conceptualization, Investigation, Methodology, Visualization, Writing – original draft, Writing – review and editing; Dominique Kranz, Investigation, Methodology, Visualization, Writing – original draft, Writing – review and editing; G Ulrich Nienhaus, Michael Boutros, Conceptualization, Funding acquisition, Investigation, Supervision, Writing – original draft, Writing – review and editing

## Author ORCIDs

Oksana Voloshanenko http://orcid.org/0000-0002-0673-3278
Dominique Kranz http://orcid.org/0000-0003-3604-7647
G Ulrich Nienhaus http://orcid.org/0000-0002-5027-3192
Michael Boutros http://orcid.org/0000-0002-9458-817X

## Decision letter and Author response

Decision letter https://doi.org/10.7554/eLife.64498.sa1
Author response https://doi.org/10.7554/eLife.64498.sa2

---

# Additional files

## Supplementary files

• Transparent reporting form
• Supplementary file 1. p-values.
• Supplementary file 2. siRNAs.
• Supplementary file 3. Antibodies.
• Supplementary file 4. Plasmids.
• Supplementary file 5. Primers for qPCR.
• Supplementary file 6. Oligonucleotides.

## Data availability

All data generated during this study is included in the manuscript.

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
