## [Editor Report]

The authors describe a fluorescent tagging strategy that allows them to knock in one copy each of differently tagged wild-type and a non-destructible β-catenin mutant, so that the behavior of the mutant can be directly compared to the wild-type protein under different activities of the β-catenin destruction complex. The differential tagging of a wild-type and mutant Wnt components offers the opportunity to compare directly their properties in a living cell, and the approach is likely to be useful for future mechanistic investigations by the Wnt and cancer communities.

---

## [Decision Letter]

**Decision letter after peer review:**

Thank you for submitting your article "Allele-specific endogenous tagging and quantitative analysis of β-catenin in colorectal cancer cells" for consideration by *eLife*. Your article has been reviewed by 3 peer reviewers, one of whom is a member of our Board of Reviewing Editors, and the evaluation has been overseen by Marianne Bronner as the Senior Editor. The reviewers have opted to remain anonymous.

The reviewers have discussed the reviews with one another and the Reviewing Editor has drafted this decision to help you prepare a revised submission.

Summary:

The authors describe a fluorescent tagging strategy that allows them to knock in one copy each of differently tagged wild-type and a non-destructible β-catenin mutant, so that the behavior of the mutant can be directly compared to the wild-type protein under different activities of the β-catenin destruction complex. The differential tagging of a wild-type and mutant Wnt components offers the opportunity to compare directly their properties in a living cell, and the approach is likely to be useful for future mechanistic investigations by the Wnt and cancer communities.

Essential revisions:

The authors show that wt and mutant β-catenins both diffuse in larger complexes that differ between wt and mutant, possibly destruction complexes vs. transcriptional partners. These are interesting observations that may ultimately provide insights into the mechanism of β-catenin destruction. There is concern regarding the apparent degradation of the cherry fluorescent protein, and although the authors rationalize what is happening based on prior literature, there is no experimental validation of these ideas. Nonetheless, the ability to IP the cherry tag and the observation of fluorescence from this tagged version is consistent with their interpretation. There are, however, a number of technical issues that require attention, specifically a more rigorous validation that the targeting is correct, and better support for the conclusions regarding nuclear β-catenin translocation.

1. The genotyping is not adequate to prove both alleles have been correctly targeted in the double targeted clone.

In the case of the double targeted cells, specific genotyping is required for the 5' and 3' ends of both alleles. As the 3' genotyping only is the same primer set for both the Cherry and Clover cassettes it is possible that only one is targeted. 3' genotyping should be shown both for the Cherry and Clover alleles separately. Especially as CRISPR was used to generate these, it is quite possible the other allele has an NHEJ generated mutation which does not amplify by PCR, meaning that the positive band is derived from just one targeted allele (without amplification from the second allele) (New Figure 1—figure supplement 2 - old Figure S1c; primer pair c).

2. Some of the clones are apparently positive in the wrong genotyping reactions.

Clone 33 and 24 are supposedly heterozygous for the Clover allele but also appear to be positive for the specific Cherry allele (New Figure 1—figure supplement 2A - figure supplement 2A - Figure S1c; primer pair b). This may indicate that the clones are carrying extra integrated copies of the targeting vectors or that the clones analysed contain a mixture of cells both of which may impact the downstream analysis.

Clone 37 also has an apparent 3rd band when PCR is performed with the internal primer set (primer pair d; Figure S1c). This may indicate a duplication of the cassette at the targeted locus. This is especially important to resolve given that the data presented does not prove both alleles are correctly targeted at the 3' end.

3. Sequencing of the targeted alleles had not been done or has not been adequately detailed

There is no real description of the sequencing done on the targeting alleles. It is not uncommon for CRISPR generated alleles to have rearrangements including nucleotide changes, indels, local rearrangements and additional integrations were also frequently generated (e.g. Codner et al. BMC Biol. 2018 Jun 21;16(1):70.). Consequently it is important that these alleles should be properly characterised.

4. Although the quantification (Figure 4) shows increase in nuclear/cytoplasmic ratio of β-catenin, the images presented do not seem to support this. Can the authors present zoomed-in images of different cells to demonstrate that this is the case?

5. There seems to be more perinuclear puncta for Cherry-β-catenin in general compared to Clover-β-catenin, and they increase in the presence of CHIR99021 compared to DMSO control. The authors do not comment on this, and how it might possibly impact the quantification of the images, given that these puncta are outside the nuclear area.

6. Consistent with point #2 above, siCLOVER experiments (Figure 5) demonstrate that a pool of Cherry-β-catenin seems to localize at the perinuclear region in cells. The authors do not comment on this and also do not provide quantification of the fluorescence imaging experiments to support their conclusion.

7. How do clover-WT and Cherry-mut-β-catenin diffuse in response to treatment with exogenous Wnt ligand, given that Wnt secretion is still required for Wnt signaling activity in HCT116 cells? Is there a difference in their nuclear translocation?

8. Why is there a band corresponding to Cherry β-catenin in the western blot of clone #33 (Figure 2)? The authors do not comment on this. Also, were all lanes loaded with equivalent amounts of total protein? The β-catenin expression levels in knock-ins are clearly higher than in the parental line.

[Editors' note: further revisions were suggested prior to acceptance, as described below.]

Thank you for submitting your article "Allele-specific endogenous tagging and quantitative analysis of β-catenin in colorectal cancer cells" for consideration by *eLife*. Your article has been reviewed by 2 peer reviewers, one of whom is a member of our Board of Reviewing Editors, and the evaluation has been overseen by Marianne Bronner as the Senior Editor. The following individual involved in review of your submission has agreed to reveal their identity: Owen J Sansom (Reviewer #3).

Essential revisions:

1. Regarding the new experiments looking at diffusion after treatment with porcupine inhibitor, which show some level of Wnt regulation of the Δ45 mutant, have the authors considered that GSK3 inhibition produced by Wnt treatment will also reduce the association of β-catenin with APC; perhaps this is the origin of their observation.

2. The discussion starting at line 430 could use some clarification. Can the authors be more specific as to what they think are the causes of the difference in diffusivity of the WT and mutant alleles? One can imagine that the mutant is bound to Axin and APC in the destruction complex and not turned over as fast as WT. The authors also mention association with transcription factors, and also cytoplasmic-nuclear shuttling of destruction complex components. Perhaps a cartoon or at least a clear delineation of the possible reasons for these behaviors would help.

3. Figure S1d is frequently referred to but is not clearly indicated. I assume it's part of S1B but not clear. These supplemental figures are also called out in the rebuttal but don't exactly correspond to what is in the manuscript. Lowercase vs. uppercase? Labeling of the individual FACS plots is needed.

---

## [Author Response]

Essential revisions:The authors show that wt and mutant β-catenins both diffuse in larger complexes that differ between wt and mutant, possibly destruction complexes vs. transcriptional partners. These are interesting observations that may ultimately provide insights into the mechanism of β-catenin destruction. There is concern regarding the apparent degradation of the cherry fluorescent protein, and although the authors rationalize what is happening based on prior literature, there is no experimental validation of these ideas. Nonetheless, the ability to IP the cherry tag and the observation of fluorescence from this tagged version is consistent with their interpretation. There are, however, a number of technical issues that require attention, specifically a more rigorous validation that the targeting is correct, and better support for the conclusions regarding nuclear β-catenin translocation.1. The genotyping is not adequate to prove both alleles have been correctly targeted in the double targeted clone.In the case of the double targeted cells, specific genotyping is required for the 5' and 3' ends of both alleles. As the 3' genotyping only is the same primer set for both the Cherry and Clover cassettes it is possible that only one is targeted. 3' genotyping should be shown both for the Cherry and Clover alleles separately. Especially as CRISPR was used to generate these, it is quite possible the other allele has an NHEJ generated mutation which does not amplify by PCR, meaning that the positive band is derived from just one targeted allele (without amplification from the second allele) (New Figure 1—figure supplement 2 - old Figure S1c; primer pair c).

To validate the correct integration of Cherry and Clover in each allele, we performed the additional experiments suggested by the reviewers (Modified Figure 1—figure supplement 2, New Figure 1—figure supplement 3,4). We now show 3’ genotyping with three different primer pairs (Modified Figure 1—figure supplement 2A primer pairs c, e, e’). While primer pair e selectively amplifies the Cherry-tagged allele and e’ selectively amplifies the Clover-tagged allele, primer set c amplifies both alleles, which can be distinguished by their size, due to the different lengths of the antibiotic resistance markers (Modified Figure 1—figure supplement 2A). All three PCRs support the correct integration of each cassette in the different clones. To make these results more accessible, we now show all primer pairs separately in Modified Figure 1—figure supplement 2A.

Furthermore, we have also performed additional genotyping of the double targeted clone 37 with primers annealing 5’ and 3’ of the integration cassettes outside of the homology arms (New Modified Figure 1—figure supplement 2B). This further supports the correct integration and does not indicate that multiple copies of the cassette are present in the target locus. Moreover, we have used the PCR amplicons spanning the modified locus for Sanger sequencing (primers are indicated in Modified Figure 1—figure supplement 2B, sequence alignments are shown in New Figure 1—figure supplement 3,4), which further supports successful gene targeting. We have revised the text to better explain these results.

Specifically, we have made the following changes to the text:

“In the following we focus on clones #24, #33, #37, #45, as they were found to represent different integration events. To confirm the editing events, genomic DNA, mRNA and protein lysates were analyzed and compared to the parental HCT116 (β-catenin^WT/∆45^) cell line and to an HCT116 isogenic cell line in which the ∆Ser45 allele was removed (β-catenin^WT+/-^) (Chan et al., 2002). First, to analyze them in depth, we used primers annealing upstream and downstream of the fluorophore tags (primers (a) in Figure 1—figure supplement 2A). While PCR amplicons of successfully tagged *CTNNB1/*β-catenin alleles are expected to be more than 3 kb in length, amplicons from the untagged locus would be approximately 1.2 kb. PCR from genomic DNA from clone #37 resulted in a single band of 3kb, indicating that both alleles of *CTNNB1* were edited. In contrast, genotyping of clones #33, #45 and #24 resulted in 3 kb and 1.2 kb amplicons, suggesting that only a single donor integrated in one allele (Figure 1-figure supplement 2A). Since genotyping with primers annealing within the donor construct cannot distinguish between homology-mediated integration at the target locus and non-targeted integration elsewhere in the genome, we next performed genotyping PCRs with one primer annealing in the *CTNNB1* locus and the other inside the integration cassette (Figure 1—figure supplement 2A) .....”

2. Some of the clones are apparently positive in the wrong genotyping reactions.Clone 33 and 24 are supposedly heterozygous for the Clover allele but also appear to be positive for the specific Cherry allele (New Figure 1—figure supplement 2A - figure supplement 2A - Figure S1c; primer pair b). This may indicate that the clones are carrying extra integrated copies of the targeting vectors or that the clones analysed contain a mixture of cells both of which may impact the downstream analysis.

We thank the reviewers for this comment. To further verify the genotype of the indicated clones, we performed additional PCRs (New primer pairs d,e and e’ – Modified Figure 1—figure supplement 2A). These include primer sets, which distinguish the Cherry and Clover alleles by their BRS/blasticidin (Cherry, new primer set e) and HGR/hygromycin (Clover, new primer set e’) resistance genes. Clones #24 and #33 result in bands with primer pair e’, but no amplification with primer pair e, indicating that they exclusively harbor a Clover-tagged allele (Modified Figure 1—figure supplement 2A). Importantly, we also performed PCR with primer pair c, where both primers anneal inside the donor plasmid and would therefore also amplify integrations elsewhere in the genome (Modified Figure 1—figure supplement 2A). Due to the size difference of the blasticidin and hygromycin resistance genes, this PCR can differentiate between both donor templates. In clones #24 and #33, we exclusively detect a larger PCR amplicon expected from the Clover donor. Together, these assays support that clones #24 and #33 only contain a single integration of the Clover donor template in one β-catenin allele.

Clone 37 also has an apparent 3rd band when PCR is performed with the internal primer set (primer pair d; Figure 1—figure supplement 2A - old Figure S1c). This may indicate a duplication of the cassette at the targeted locus. This is especially important to resolve given that the data presented does not prove both alleles are correctly targeted at the 3' end.

As described above, we performed genotyping with a number of additional primer pairs (New primer pairs d,e and e’ - Modified Figure 1—figure supplement 2A, New Figure 1—figure supplement 2B). We would expect to detect larger bands in the case of full or partial cassette duplication by most of the primers (Modified Figure 1—figure supplement 2A, New Figure 1—figure supplement 2B). We do not detect such bands, however, indicating single copy integration of our donor sequences. Occasionally, a weak and often fuzzy DNA signal appears above the main bands, which we interpret as an unspecific amplification event or artefact of the agarose gels.

3. Sequencing of the targeted alleles had not been done or has not been adequately detailed.There is no real description of the sequencing done on the targeting alleles. It is not uncommon for CRISPR generated alleles to have rearrangements including nucleotide changes, indels, local rearrangements and additional integrations were also frequently generated (e.g. Codner et al. BMC Biol. 2018 Jun 21;16(1):70.). Consequently it is important that these alleles should be properly characterised.

We thank the reviewers for highlighting that the sequencing results were not adequately discussed in the initial submission. We sequenced the whole set of clones and the results are summarized in the table presented as Figure S1C. As expected, we found that several clones carried mutations and/or rearrangements (Figure S1C). Sequencing also confirmed that only clone #37, which has been used for the bi-allelic comparison presented in this manuscript, harbors an integration cassette in both alleles. Importantly, extended sequence analysis of clone #37 covered both integrated regions completely and confirmed correct integration (Modified Figure 1—figure supplement 2A, New Figure 1—figure supplement 2,3,4).

4. Although the quantification (Figure 4) shows increase in nuclear/cytoplasmic ratio of β-catenin, the images presented do not seem to support this. Can the authors present zoomed-in images of different cells to demonstrate that this is the case?

To clarify this point we have added enlarged images and show them in the New Figure 4—figure supplement 2.

5. There seems to be more perinuclear puncta for Cherry-β-catenin in general compared to Clover-β-catenin, and they increase in the presence of CHIR99021 compared to DMSO control. The authors do not comment on this, and how it might possibly impact the quantification of the images, given that these puncta are outside the nuclear area.

As suggested by the reviewer, we looked specifically for perinuclear puncta of both alleles of βcatenin and detected them for both WT and mutated alleles. We did not quantify them, since this was not the focus of our study, but we did not notice a significant difference. We could detect cells without these puncta, just puncta with one color (WT or mutant allele) or yellow (WT and mutant allele together in the same puncta). Similar structures were described upon overexpression of the amino-terminal part of β-catenin by Giannini et al. (2000).

**Author response image 1. sa2fig1:** Both alleles, β-catenin^WTClover^ and β-catenin^Δ45Cherry^, can be located in the perinuclear puncta. Representative immunofluorescence images of HCT116 β-catenin^wtClover/Δ45Cherry^ cells from experiments shown in Figure 4C, 5B. Scale bar: 10 μm.

6. Consistent with point #2 above, siCLOVER experiments (Figure 5) demonstrate that a pool of Cherry-β-catenin seems to localize at the perinuclear region in cells. The authors do not comment on this and also do not provide quantification of the fluorescence imaging experiments to support their conclusion.

As suggested by the reviewer, we quantified immunofluorescence as shown in New Figure 5B, New Figure 5—figure supplement 1. Since β-catenin is difficult to detect on an endogenous level, its quantification is challenging, especially upon downregulation, and interpretation of the results is limited. New Figure 5B represents one of the four biologically independent experiments and normalized results of four independent ones are provided in New Figure 5—figure supplement 1.

7. How do clover-WT and Cherry-mut-β-catenin diffuse in response to treatment with exogenous Wnt ligand, given that Wnt secretion is still required for Wnt signaling activity in HCT116 cells? Is there a difference in their nuclear translocation?

As indicated by the reviewer, HCT116 cells express canonical Wnt ligands, resulting in strong Wnt pathway activity. Therefore, it is not surprising that no changes were detected by IF upon addition of recombinant Wnt3a (New Figure 4C). Presumably, the cells reached maximal Wnt activity even without Wnt ligand stimulation, as shown previously (Voloshanenko et al. Nature Commun 2013). Hence, to address the role of Wnt secretion, we inhibited Wnt protein secretion using the porcupine inhibitor LGK974. First, we treated HCT116 #37 cells with the porcupine inhibitor LGK974 for five to six days to inhibit endogenous Wnt ligand secretion, then added 200 ng/ml of recombinant Wnt3a to induce Wnt signaling and performed immunofluorescence analysis 16 h later. Upon inhibition of Wnt secretion from HCT116 #37 cells, we observed a strong decrease of β-catenin that was rescued upon addition of recombinant Wnt3a (New Figure 4C). Surprisingly, not only β-catenin-Clover WT protein but also β-catenin-Cherry ∆Ser45 protein was regulated by LGK974. As we saw striking differences in the signal of β-catenin, we cannot measure translocation from the cytoplasm to the nucleus of the protein.

The quantification of IF data of endogenous β-catenin especially upon signal reduction could lead to an over-interpretation of the data as the signal intensity is not linear. To confirm this result, we performed FCS measurements of β-catenin alleles after LGK794 treatment. Indeed, concentrations of both alleles of β-catenin are regulated by LGK974 treatment. β-catenin-Clover WT protein concentration in the cytoplasm is strongly dependent on Wnt secretion, for which the concentration in the cytosol was reduced from C_WT,LGK_ = 36 ± 10 nM to C_WT,LGK_ = 7.2 ± 2.6 nM. Additional treatment with recombinant Wnt3a raised the concentration again and almost reversed the effects of the LGK treatment (C_WT,LGK_^+^_Wnt3a_ = 20 ± 8 nM, Cytosol). The concentration of the mutant β-catenin-Cherry decreased from C_Δ45,DMSO_ = 101 ± 18 nM to C_Δ45,LGK_ = 79 ± 13 nM upon LGK974 treatment and rose again to C_Δ45,LGK_^+^_Wnt3a_ = 94 ± 22 nM by adding Wnt3a (New Figure 6D, New Figure 6—figure supplement 1B).

8. Why is there a band corresponding to Cherry β-catenin in the western blot of clone #33 (Figure 2)? The authors do not comment on this. Also, were all lanes loaded with equivalent amounts of total protein? The β-catenin expression levels in knock-ins are clearly higher than in the parental line.

To improve the presentation of Figure 2B, we now indicated the different isoforms of β-catenin in the Western blot. Regarding clone #33, the lower band corresponds to the WT allele of β-catenin indicating the heterozygosity of clone #33. β-actin serves as a loading control in Figure 2B. Since less total protein was detected in the lane of the parental cell line, this might explain the lower β-catenin levels. Comparison of qPCR (Figure 3B) and CTNNB1/β-catenin mRNA expression levels indicated no differences between parental cells and clones. To clarify this point, we made additional Western blots to compare the parental cells and clone #37 and added them as New Figure 2—figure supplement 1A.

[Editors' note: further revisions were suggested prior to acceptance, as described below.]

Essential revisions:1. Regarding the new experiments looking at diffusion after treatment with porcupine inhibitor, which show some level of Wnt regulation of the Δ45 mutant, have the authors considered that GSK3 inhibition produced by Wnt treatment will also reduce the association of β-catenin with APC; perhaps this is the origin of their observation.

Our results indicate that the mutant form of β-catenin can also (in part) be regulated by Wnt signaling, likely involving the regulation of its interaction with APC. In line with our report, previous studies have indicated that β-catenin mutants with deletion or a mutation of Ser 45 retain signaling activity and are still responsive towards Wnt activation (PMID: 12060769 and PMID: 32751567). Together, these data suggest that, despite the absence of Ser45 phosphorylation, interactions of β-catenin with the destruction complex and its proteasomal degradation still occur. Further studies will be required to test if inhibition of GSK3 after Wnt stimulation is an essential mechanism that controls the cellular behavior of mutant β-catenin, including its interaction with APC.

2. The discussion starting at line 430 could use some clarification. Can the authors be more specific as to what they think are the causes of the difference in diffusivity of the WT and mutant alleles? One can imagine that the mutant is bound to Axin and APC in the destruction complex and not turned over as fast as WT. The authors also mention association with transcription factors, and also cytoplasmic-nuclear shuttling of destruction complex components. Perhaps a cartoon or at least a clear delineation of the possible reasons for these behaviors would help.

As suggested by the reviewers, we have rephrased and expanded this part of the discussion to indicate the different possibilities. In short, wt β-catenin protein diffuses faster than mutant β-catenin, suggesting that wt protein is present as individual molecules or as part of smaller complexes, whereas the mutant is part of larger, more slowly diffusing complexes. FCS analysis only reveals the size but not the composition of the complexes. Hence, we can only speculate about the different possibilities. For example, in the nucleus, the mutant form presumably already interacts with the Wnt enhanceosome, in the absence of stimulation. We agree with the reviewer that, in the cytoplasm, the mutant form is likely to interact with the destruction complex, but might not be turned over as fast as the wt isoform. In the revised manuscript, we discuss these possibilities in much more detail.

3. Figure S1d is frequently referred to but is not clearly indicated. I assume it's part of S1B but not clear. These supplemental figures are also called out in the rebuttal but don't exactly correspond to what is in the manuscript. Lowercase vs. uppercase? Labeling of the individual FACS plots is needed.

We would like to thank the reviewer for pointing out the inconsistency in the figure labels, which we have corrected in the revised manuscript. We have added the labeling of the FACS plots, which was accidentally removed from the Figure 1—figure supplement 1B. Now every individual plot is labeled as (i), (ii), (iii) and (iv). We have also corrected the inconsistencies in the figure labeling.